



# Models of buoyancy-driven dykes using continuum plasticity and fracture mechanics: a comparison

Yuan Li[1], Timothy Davis[1,2], Adina E. Pusok[1], and Richard F. Katz[1]

[1]Department of Earth Sciences, University of Oxford, Oxford, UK
[2]School of Earth Sciences, University of Bristol, Bristol, UK

**Correspondence:** Yuan Li (Yuan.Li@earth.ox.ac.uk)

**Abstract.** Magmatic dykes are thought to play an important role in the thermomechanics of tectonic rifting of the lithosphere. Our understanding of this role is limited by the lack of models that consistently capture the interaction between magmatism, including dyking, and tectonic deformation. While linear elastic fracture mechanics (LEFM) has provided a basis for understanding the mechanics of dykes, it is difficult to consistently incorporate LEFM into geodynamic models. Here we further

develop a continuum theory that represents dykes as plastic tensile failure in a two-phase, Stokes–Darcy model with a poro-viscoelastic–viscoplastic (poro-VEVP) rheological law (Li et al., 2023). We validate this approach by making quantitative comparison with LEFM, enabled by a novel poro-LEFM formulation. The comparison shows that dykes in our continuum theory propagate slowly—a consequence of Darcian drag on the magma. Moreover, dissipation of mechanical energy in the poro-VEVP model implies a high critical stress intensity in LEFM. We improve the poro-VEVP model by reformulating the

compaction stress and incorporating anisotropic permeability in regions of plastic failure.

## 1   Introduction

Magmatic dykes, formed by fluid-driven fracture, are important pathways for magma ascent across the lithosphere. This is particularly true at rift zones, where they are promoted by both magma supply and tectonic extension (Buck, 2006). Dykes may reach the surface and fuel volcanic eruptions, or may stall and solidify at depth within the crust (Fiske et al., 1997; Gudmunds-

son and Loetveit, 2005; Delcamp et al., 2012; Passarelli et al., 2014; Maccaferri et al., 2014). Dyke propagation is affected by the ambient stress field comprising tectonic stress, topographic loading (McGuire and Pullen, 1989; Fernández et al., 2002; Maccaferri et al., 2014; Rivalta et al., 2015; Sigmundsson et al., 2024), and crustal heterogeneity (Thiele et al., 2020; Drymoni et al., 2023). However, dyke propagation can also modify the ambient stress field and weaken the lithosphere (Kjøll et al., 2019; Brune et al., 2023). Consistently incorporating dyking in geodynamic models is therefore crucial for understanding rifting pro-

cesses; this remains an outstanding challenge. Here we describe progress in developing and validating an approach whereby dyking is modelled as plastic failure in a continuum, two-phase theory for partially molten rock.

In most previous work, the mechanics of dykes is formulated in terms of linear elastic fracture mechanics (LEFM). LEFM conceptualises dykes as mode-I fractures opened at the tip and widened by magma flow (Rivalta et al., 2015). The magmatic flow is modelled as viscous and parallel, in the narrow gap between the dyke walls, as shown in the schematic in Fig. 1(a). The





gap opens behind a sharp tip, where elastic stress in the wall rock overcomes the fracture toughness and promotes tip advance. The elastic stress arises from a combination of the pressure of fluid within the dyke and the preexisting stress field surrounding it.

    LEFM models have explored the propagation rate and geometry of two-dimensional fractures with constant flux (Lister, 1990; Roper and Lister, 2007), as well as two- and three-dimensional fractures with constant volume (Spence and Turcotte,

1990; Davis et al., 2020, 2023). These magmatic fractures can be slowed or arrested due to loss of volatiles and heat, and by solidification (Rubin, 1995; McLeod and Tait, 1999; Bolchover and Lister, 1999; Taisne et al., 2011; Rivalta et al., 2015; Abdullin et al., 2024). The direction of propagation has been investigated in relation to tectonic stress, topographic loading, and crustal heterogeneity (Maccaferri et al., 2014; Acocella et al., 2024). Despite the many successes of the LEFM approach, there are significant obstacles to consistently embedding it into models that account for the causes, dynamics and consequences

of dyking.

    In the geodynamic context of the hot, ductile asthenosphere, magma transport has long been modelled using a poro-viscous, Stokes–Darcy theory (McKenzie, 1984; Katz, 2022). This two-phase continuum formulation has been applied to geological settings including mid-ocean ridges (e.g., Sim et al., 2020; Pusok et al., 2022b), subduction zones (e.g., Rees Jones et al., 2018; Cerpa et al., 2018), and beneath continents (e.g., Schmeling et al., 2019). These studies were limited to hot asthenospheric

regions by the use of a purely viscous rheological law.

    In other work, the theory has been extended to accommodate elastic and brittle deformation at lower temperatures. This extension aimed to model melt transport upward across the ductile–brittle transition. Notably, Keller et al. (2013) first incorporated plastic failure into a two-phase continuum model of magmatism. Li et al. (2023) improved the theoretical formulation by employing a poro-viscoelastic–viscoplastic (poro-VEVP) rheology (Duretz et al., 2021) and proposing a new hyperbolic yield

surface to address physical, mathematical and computational issues of Keller's model. Li et al. (2023) showed how dyke-like features emerge from this formulation and bear a quantitative similarity with dykes described by LEFM theory. In particular, Li et al. (2023) observed that a poro-VEVP dyke can be narrow and fast relative to advection and (de)compaction in poro-viscous dynamics (Katz et al., 2022), and the stress distribution around its tip matches the LEFM model for some value of critical stress intensity.

However, further validation and exploration of the capabilities of the continuum representation of dykes are necessary. Key differences with LEFM theory are readily noted: Darcian versus Poiseuille flow of the liquid phase; plastic yield versus brittle fracture of the solid phase. In this comparison, two major issues require further investigation. The first is the slower propagation speed predicted by the poro-VEVP formulation ($\sim$1 m/yr versus $\sim$1 km/day (Davis et al., 2023)). The second is the very high critical stress intensity needed in LEFM for consistency between the predictions ($\sim$1.5 GPa m$^{1/2}$). The previous benchmark in

Li et al. (2023) is also incomplete in that the poro-VEVP dyke was driven by far-field tensile stresses, not buoyancy, and did not reproduce the classic LEFM cases of constant-flux or constant-volume for comparison.

    To organise our investigation of these issues, we propose two hypotheses. We hypothesise that the slow speed of poro-VEVP dyke propagation is due to the greater viscous resistance to magma ascent in Darcian porous flow compared to Poiseuille flow. Furthermore, we hypothesise that the fracture toughness that provides an equivalent resistance to dyke propagation can be di-





rectly calculated from the rate of plastic energy dissipation in the poro-VEVP model. We verify these hypotheses by simulating a constant-flux, buoyancy-driven fracture in the poro-VEVP model and making quantitative comparison to a corresponding LEFM model. Our simulations are set up such that poro-VEVP fracture propagates vertically at a constant speed.

To facilitate the comparison, we introduce a modified LEFM model in which the interior of the dyke is a porous medium. This assumes a dyke region with fixed width but variable porosity (Fig. 1(b)). In this poro-LEFM model, Darcy flow supplies buoyant fluid to a toughness-dominated tip embedded in an elastic medium. The poro-LEFM model converges to the classical LEFM model in the limit of the porosity going to unity. However, at smaller porosity, it facilitates a direct comparison with the poro-VEVP model in terms of stress distribution, porosity profile and dyke propagation speed. We show that through the use of a poro-LEFM fracture toughness, calculated with a poro-VEVP energy analysis, there is a good match between the two models. This establishes a physics-based, quantitative relation between the poro-VEVP and LEFM models. Moreover, it advances our understanding of how distributed plastic failure affects dyke propagation.

As we detail below, this comparison also highlights a shortcoming of the poro-VEVP model. Isotropic permeability within the poro-VEVP dyke promotes widening by horizontal porous flow, a behaviour not associated with real (or LEFM) dykes. We resolve this discrepancy by introducing an anisotropic permeability tensor into the poro-VEVP model to limit leakage and enhance fracture propagation (e.g. Snow, 1969). Anisotropic permeability can arise from anisotropic stresses and aligned pores or fractures (e.g. Snow, 1969; Sibson, 1996; Daines and Kohlstedt, 1997; Li et al., 2009; Takei, 2010; Taylor-West and Katz, 2015; Lang et al., 2018; Seltzer et al., 2023; Bader et al., 2024).

This manuscript is organised as follows. The next section (Sec. 2) develops the poro-LEFM model, details the poro-VEVP model, and explains how energy dissipation is used to evaluate fracture toughness. The results section (Sec. 3) illustrates the steadily propagating dykes produced by the poro-VEVP model. The results section also verifies the estimated toughness by comparing poro-VEVP and poro-LEFM models in terms of their porosity and stress distributions. We discuss the results and their broader relevance in section 4 and summarise in section 5.

## 2 Models of a buoyancy-driven dyke

In this section, we develop two distinct models that are both aimed to describe buoyancy-driven dyke ascent. We first introduce the poro-LEFM model, which differs from the standard LEFM model in that it treats the dyke interior as a porous medium. We then review the continuum mechanical, poro-VEVP model developed in Li et al. (2023), and equip it with two key enhancements: a reformulated compaction pressure for improved numerical robustness and anisotropic permeability to impose a preferred dyke-parallel direction of Darcy flow. Finally, we develop an analysis of mechanical energy dissipation in the poro-VEVP representation of a dyke. This provides a quantitative estimate of the effective fracture toughness for the poro-LEFM model, and hence a basis for comparing the models.





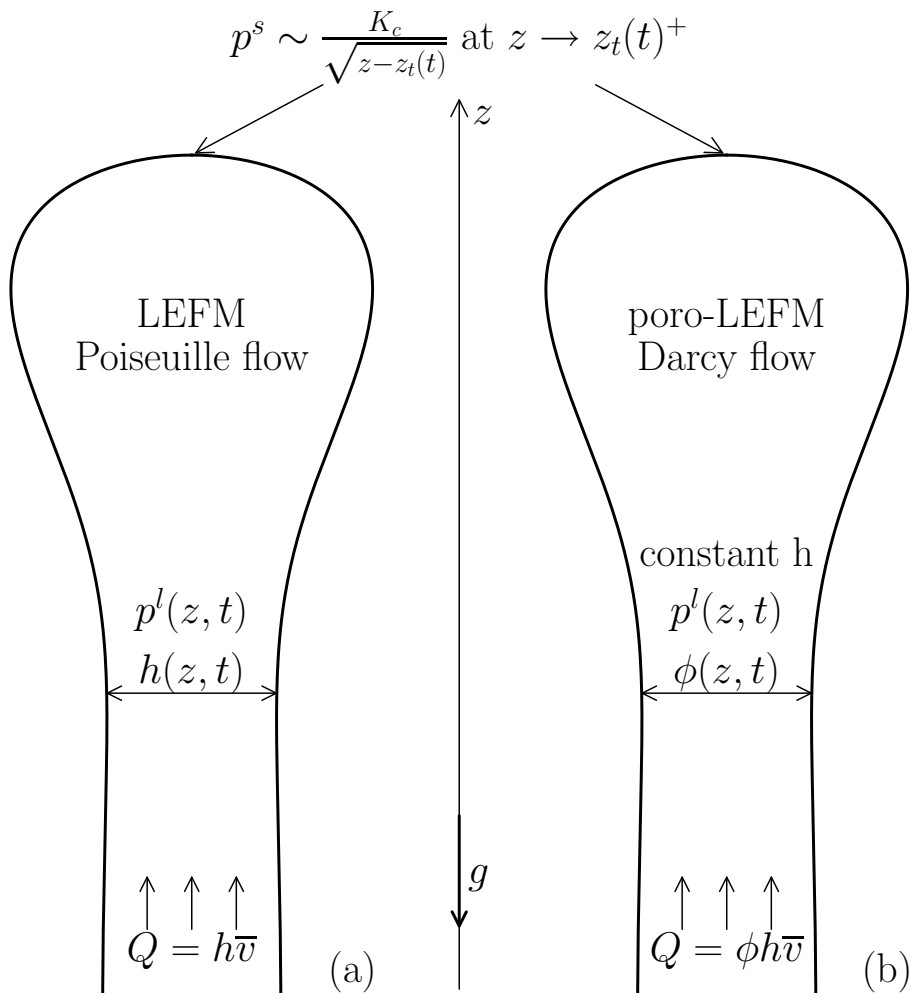

**Figure 1.** Sketch of the LEFM (e.g. Lister, 1990) and poro-LEFM models for a buoyancy-driven fracture. Here, $Q$ denotes the volume flux through the fracture and $\bar{v}$ represents the cross-section average of vertical velocity component of the liquid. Both $Q$ and $\bar{v}$ are constants at $z \to -\infty$. The far-field conditions and the definition of other notations are presented in section 2.



## 2.1 The poro-LEFM formulation

The development of the poro-LEFM model follows Lister (1990) both conceptually and mathematically. This section gives an overview; full details are available in Appendix A.

Similar to the classic LEFM model in Figure 1(a), we consider a vertical, two-dimensional channel as shown in Figure 1(b), extending from $-\infty$ to a tip at position $z = z_t$. This channel represents an idealised dyke where buoyant fluid flows upward, deforms the elastic solid phase, and drives the fracture at the tip. Along the fracture walls, the elastic normal stress $p^s$ is intensified by a critical factor $K_c$ near the tip.

Unlike the LEFM model, which assumes Poiseuille flow in an open channel of variable width, the poro-LEFM model assumes porous flow in a permeable channel of uniform, fixed width $h$ and variable porosity $\phi(z,t)$. The porous flow is modulated by a porosity-dependent mobility $M_\phi = k_\phi/\mu$, where $k_\phi$ is the permeability and $\mu$ is the liquid viscosity. We assume this porous flow is driven purely by buoyancy, leading to a constant porosity $\phi_0$ in the tail region, which we refer to as the far field.

The mathematical formulation includes Darcy's law for the liquid flux $\phi v$, an elastic-stress balance equation, and boundary conditions at the tip and the far field,

$$\phi v = M_\phi \left( -\frac{\partial p^l}{\partial z} + \Delta\rho g \right), \tag{1}$$

$$p^s(z,t) = -\left( \frac{G}{1-\nu} \right) \frac{1}{2\pi} \int_{-\infty}^{\infty} \frac{\partial h\phi(\xi,t)}{\partial \xi} \frac{\mathrm{d}\xi}{\xi - z}, \tag{2}$$

$$p^s(z,t) \approx -\frac{K_c}{[2(z-z_t)]^{1/2}}, \quad \text{at } z \to z_t^+, \tag{3}$$

$$\phi \approx \phi_0, \text{ at } z \to -\infty. \tag{4}$$

Here $v$ is the vertical component of liquid velocity, $p^l$ is the dynamic liquid pressure (assumed equal to $p^s$ inside the dyke), $\Delta\rho = \rho^s - \rho^l$ is the density difference between solid ($s$) and liquid ($l$), $g$ is the gravitational force per unit mass, $G$ and $\nu$ are the elastic shear modulus and Poisson's ratio of the solid, and $K_c$ is the critical stress intensity. In this manuscript we select $\nu = 1/2$, which enforces that the solid phase is incompressible.

For a purely buoyancy-driven flow, buoyancy is balanced by the Darcian drag force and the propagation speed $c$ becomes constant. This speed is obtained by solving Eq. (1) for $v$ as $z \to -\infty$, where there is no gradient of dynamic pressure,

$$c = \frac{M_\phi(\phi_0)}{\phi_0} \Delta\rho g. \tag{5}$$

Here $M_\phi(\phi_0)$ is the fluid mobility at $\phi = \phi_0$. This implies a constant volume flux from the far field, $Q_0 \equiv \phi_0 h c$. We compare this result with a canonical LEFM, buoyancy-driven, open fracture having far-field width $h_0$ and hence tip speed $c_f = h_0^2 \Delta\rho g/12\mu$ (Lister, 1990).





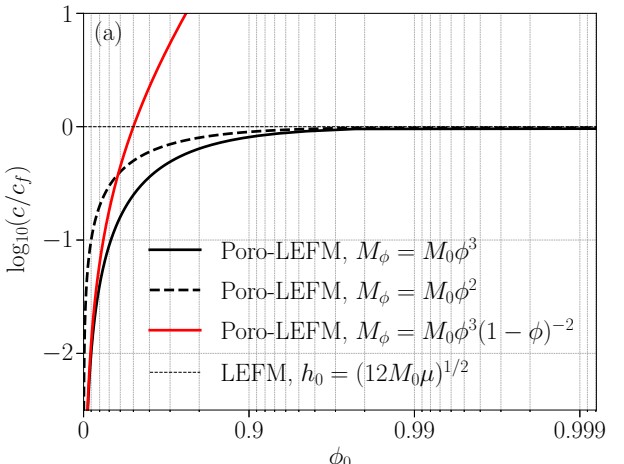
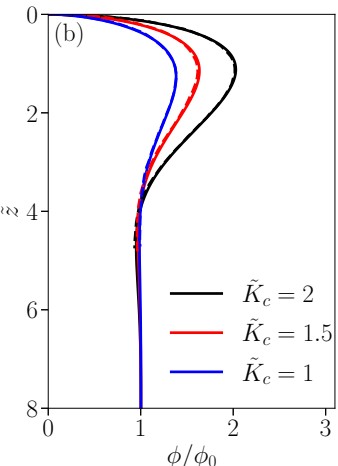

**Figure 2.** Comparison of LEFM and poro-LEFM models. (a) Tip propagation speed as a function of melt fraction. The horizontal axis is presented in a logarithmic scale. Three different permeability–porosity relationships are considered in the poro-LEFM model. The mobility prefactor $M_0$ is defined as $M_0 = h_0^2/12\mu$, ensuring that $c \to c_f$ when $\phi_0 \to 1$. (b) Profiles of porosity at different values of $\tilde{K}_c$ in the poro-LEFM model (solid lines) compared with the profiles of scaled fracture width $h/h_0$ in the LEFM model (dashed lines) (Lister, 1990, Fig. 3).Here, $\tilde{K}_c$ and $\tilde{z}$ denote scaled $K_c$ and $z$, respectively.

Figure 2(a) shows how the poro-LEFM steady propagation speed $c$ increases with the far-field porosity $\phi_0$ for two choices of fluid mobility: a power-law relation $M_\phi = M_0\phi^n$, where $n = 2$ and $3$, and the Kozeny-Carman relation $M_\phi = M_0\phi^3(1-\phi)^{-2}$.

We choose $M_0 = h_0^2/12\mu$ to achieve a convergence between $c$ and $c_f$. In particular, with our choice of $M_0$ in the power-law permeability relation, the speed $c$ approaches $c_f$ as $\phi_0 \to 1$. We adopt the cubic porosity dependence for the remainder of this manuscript to enforce a quantitative relationship between poro-LEFM and canonical LEFM theory.

We solve the system of equations and boundary conditions (1)–(4) after rescaling variables and transforming into a coordinate system that moves with the tip (see Appendix A for details). Solutions for $\phi(z)$ are obtained with the numerical procedure

given by Roper and Lister (2007).

Figure 2(b) presents results for three choices of $K_c$. The porosity is non-dimensionalised by the far-field porosity $\phi_0$. All porosity profiles show a bulging head approaching the tip at which $\phi = 0$, and a constant value in the tail where $\phi = \phi_0$. The head widens (again, in terms of the porosity) with increasing $K_c$, giving a larger solid deformation and therefore reflecting the increasing stress required to propagate the tip.

Figure 2 verifies the anticipated alignment between the poro-LEFM and LEFM models. Panel (a) displays the convergence of propagation speeds when $M_0$ is judiciously selected as noted above. It is important to recognise that for the far-field volume flux to converge as $\phi \to 1$, the poro-LEFM width must equal the far-field width of the LEFM dyke, meaning $h = h_0$. Panel (b) shows the quantitative equivalence between the porosity distribution in a poro-LEFM dyke and the width variation in an





LEFM dyke in dimensionless terms, corroborated by the numerical results from Lister (1990). This equivalence is also clear
by comparing the dimensionless equations Eqs. (A10)–(A13) in Appendix A with Eqs. (2.8)–(2.10) of Lister (1990).

## 2.2   The poro-viscoelastic–viscoplastic (poro-VEVP) formulation

This section presents a two-dimensional (2-D) Stokes–Darcy model for simulating a buoyancy-driven dyke with constant liquid
influx from the boundary. This model shares Darcy's equation and mass continuity equation with the poro-LEFM model, but
in 2-D form and taking into account the solid velocity. The stress-balance equation for the solid phase is more complex, bal-
ancing stresses of the two-phase medium in the context of a poro-viscoelastic–viscoplastic (poro-VEVP) rheological law. The
solid phase deforms as a Maxwell material combining viscous, elastic, and viscoplastic elements, with a Kelvin viscosity for
regularisation of plasticity. For more details on this poro-VEVP model, see Li et al. (2023). Here, we focus on improvements to
the poro-VEVP model for simulating a constant-width, fluid-driven fracture in a porous medium and explain the computational
model setup.

### 2.2.1   Stress-balance equation and a new compaction formulation

Stress-balance of a two-phase medium satisfies

$$-\boldsymbol{\nabla}p^l + \boldsymbol{\nabla}\cdot\left[(1-\phi)\boldsymbol{\tau}^s\right] - \boldsymbol{\nabla}\left[(1-\phi)\Delta P\right] - \phi\Delta\rho\boldsymbol{g} = \boldsymbol{0}. \tag{6}$$

Here, $(1-\phi)\boldsymbol{\tau}^s$ and $-(1-\phi)\Delta P$ represent the effective shear and decompaction stresses, respectively. $\Delta P = p^s - p^l$ is the
pressure difference between phases. The shear and decompaction stresses must be expressed in terms of strain rates and must
also be constrained by the plastic yield condition. This challenge was addressed by Li et al. (2023) and we follow their approach,
with a small modification.

Previous studies employed an effective-viscosity method for both shear and compaction (e.g., Moresi et al., 2003; Keller
et al., 2013; Li et al., 2023). While this approach is appropriate for shear, it can lead to a divergence of the effective compaction
viscosity during plastic failure, compromising computational robustness (Appendix B). We propose a new formulation of $\Delta P$
to resolve this, which compares with the old formulation as follows,

$$\text{Old formulation:} \qquad (1-\phi)\Delta P = -\zeta_{\text{eff}}\mathcal{C}', \tag{7}$$

$$\text{New formulation:} \qquad (1-\phi)\Delta P = -\zeta^{\text{ve}}\mathcal{C}' + (1-\phi)\Delta P_{dl}, \tag{8}$$

where

$$\mathcal{C}' = \left[\mathcal{C} - \frac{(1-\phi)\Delta P^o}{Z_\phi\Delta t}\right], \quad \zeta^{\text{ve}} = \left(\frac{1}{\zeta_\phi^{\text{v}}} + \frac{1}{Z_\phi\Delta t}\right)^{-1}. \tag{9}$$



Here, $\mathcal{C}$ is the solid decompaction rate, $\zeta_\phi^{\mathrm{v}}$ and $Z_\phi$ are the compaction viscosity and bulk modulus, $\Delta t$ is the time step, and $\Delta P^o$ is the overpressure in previous time step. Both the effective viscosity $\zeta_{\mathrm{eff}}$ in the old formulation and the dilatancy pressure $(1-\phi)\Delta P_{dl}$ in the new formulation are parameters utilised to enforce the plastic yielding limit in the stress-balance equation.

     When the plastic yield limit is not reached, $\zeta_{\mathrm{eff}} = \zeta^{\mathrm{ve}}$ and $\Delta P_{dl} = 0$, making the two formulations equivalent. During plastic yield, $\Delta P$ is calculated from the plastic model, and either formulation can be rearranged to obtain the corresponding parameter

while maintaining a fixed $\mathcal{C}'$. The old formulation calculates the effective compaction viscosity as $\zeta_{\mathrm{eff}} = (1-\phi)\Delta P/\mathcal{C}'$ and feeds $\zeta_{\mathrm{eff}}$ to the stress-balance equation as a constant, which becomes infinity when $\mathcal{C}' = 0$. This infinite $\zeta_{\mathrm{eff}}$ impacts the convergence of the solver for the velocity field from the stress-balance equation. The new formulation resolves this issue by calculating $(1-\phi)\Delta P_{dl} = (1-\phi)\Delta P + \zeta^{\mathrm{ve}}\mathcal{C}'$ instead, and feeding it to the stress-balance solver as a constant. This approach only changes how the stress-balance equation is linked with the plastic yield condition, without altering either of the two

physics. The new constant always remains finite, improving the robustness of the computational codes.

### 2.2.2    Anisotropic permeability due to plastic failure

In the poro-VEVP model with isotropic permeability, a vertical, porous dyke inevitably widens over time due to liquid flux across the walls of the dyke. However, for consistency with the buoyancy-driven poro-LEFM model in Section 2.1, a constant width is essential. Therefore, to limit leakage through the walls, we introduce an anisotropic permeability (Snow, 1969). This

is used to ensure that the poro-VEVP dyke's width can remain constant over time, even as porosity within it may vary due to (de)compaction.

     Anisotropic permeability can be thought of as a macroscopic representation of melt-preferred orientation (MPO), which refers to the alignment of interconnected, melt-filled pores at grain scale in partially molten rocks (e.g. Daines and Kohlstedt, 1997; Takei, 2010; Bader et al., 2024). Under the effect of differential stresses, these pores align and elongate perpendicular to

the direction of maximum tension, causing differences in fluid transmissivity in different directions.

     Mode-I fractures in a porous medium, from grain-scale microcrack damage to fractures that span large numbers of grains, have an effect on liquid permeability that is similar to MPO. They create anisotropic permeability that favours flow along the fracture. Indeed, macroscopic Mode-I fractures have been conceptualised as the result of the propagation of microcracks under tension, with the propagation direction perpendicular to the direction of maximum tension (e.g. Griffith, 1921; Murrell, 1964).

Aligned microcracks are closely analogous to aligned, elongated pores. We therefore assume that mode-I fractures also cause an anisotropic permeability.

     To incorporate permeability anisotropy, we use a rank-2 tensor $\boldsymbol{M}_\phi$ to express the liquid mobility, with a size matching the problem's dimensionality. Darcy's equation is then written as

$$\phi(\boldsymbol{v}^l - \boldsymbol{v}^s) = -\boldsymbol{M}_\phi \cdot \left(\boldsymbol{\nabla} p^l + \Delta\rho\boldsymbol{g}\right), \quad \text{where} \quad \boldsymbol{M}_\phi = M_0\phi^n\boldsymbol{M}_a. \tag{10}$$

Here, $\boldsymbol{v}^l$ and $\boldsymbol{v}^s$ represent liquid and solid velocity, respectively. $\boldsymbol{M}_a$ represents the anisotropic modification. When $\boldsymbol{M}_a$ is the identity tensor, the mobility is isotropic and the equation above becomes the standard Darcy's equation.





For vertically propagating dykes simulated in this manuscript, $M_a$ is a diagonal matrix,

$$M_a = \begin{pmatrix} k_{xx} & 0 \\ 0 & k_{zz} \end{pmatrix}, \quad \text{where} \quad k_{xx}, k_{zz} \in (0, k_a). \tag{11}$$

Here, $k_a$ is a prescribed maximum permeability enhancement. We define $k_{xx}$ and $k_{zz}$ based on the plastic strain components,
$e_{xx}^{\mathrm{K}}$ and $e_{zz}^{\mathrm{K}}$, for example,

$$k_{xx} = \frac{k_a}{1 + (k_a - 1)\exp(r_{xx}/\epsilon)}, \quad \text{with} \quad r_{xx} = \frac{e_{xx}^{\mathrm{K}}}{e_{xx}^{\mathrm{K}} + e_{zz}^{\mathrm{K}}} - \frac{1}{2}. \tag{12}$$

Similarly, $k_{zz}$ is defined in terms of $r_{zz}$, which is written by replacing $e_{xx}^{\mathrm{K}}$ with $e_{zz}^{\mathrm{K}}$ in the numerator of $r_{xx}$. In equation (12) and its variant for $k_{zz}$, the quantities $r_{xx}$ and $r_{zz}$ measure the anisotropy of accumulated plastic strain in the $x$- and $z$-directions, respectively. Both are equal to $0$ when $e_{xx}^{\mathrm{K}} = e_{zz}^{\mathrm{K}}$, leading to $k_{xx} = k_{zz} = 1$, indicating isotropic mobility. The anisotropy of
mobility is related to the anisotropy of plastic strain by $\epsilon \sim 5\%$, a characteristic scale of strain anisotropy. As we model only small deformations in this manuscript, we neglect advection of plastic strains.

### 2.2.3 Rheological parameters

To facilitate comparison with the poro-LEFM model, we aim to align the rheology of the poro-VEVP model as closely as possible. Moreover, our focus here is on relating plastic deformation in a two-phase continuum to fluid-driven fracture. Therefore we
suppress viscous deformation by assigning effectively infinite values to both the shear and compaction viscosity. Furthermore, we assign a relative small, constant value to the Kelvin viscoplastic viscosity $\eta^{\mathrm{K}}$. The impact of this viscosity is discussed in section 2.3.

The elastic shear ($G_\phi$) and bulk ($Z_\phi$) moduli follow porosity-dependent relations,

$$G_\phi = (1 - \phi)G, \quad Z_\phi = (1 - \phi)Z. \tag{13}$$

Here $G$, and $Z$ are constant, reference values. Note that this bulk modulus relates to the compaction of a solid–liquid aggregate, not to the compressibility of the solid phase. In fact, we assume that the solid phase is incompressible, which is enforced in the mass conservation equation.

### 2.2.4 Computational model

The model formulation closely follows that of Li et al. (2023) with the modifications discussed above. Appendix C reviews
the full system of equations. We solve the momentum and mass conservation equations using the FD-PDE framework (Pusok et al., 2022a), built on PETSc (Balay et al., 2022a). The model domain $\Omega$ is a tall rectangle, 2.44 km in width and 20 km in height. It is discretised using a $61 \times 500$ grid with a cell size of $\Delta x = \Delta z = 40$ m. We refer to the bottom boundary as $B$.





A short time-step of $\Delta t = 1$ yr is chosen to ensure solution accuracy. This time-step is reduced further when the maximum permeability enhancement ($k_a$) increases. Details are discussed in section 3.1.

The initial porosity field has a maximum value of 0.2 at the centre of $B$. The initial porosity decays laterally with a length scale of $10^{-4}$ km and vertically with scale 0.8 km according to a Gaussian function. This implies an initial width of one cell (40 m).

To exclude the effect of external forces on the solution within the domain, we prescribe zero shear and normal stresses on all boundaries except the bottom. Along $B$ we prescribe zero shear stress and zero normal velocity of the solid phase. Liquid

flows across $B$ at a constant volume rate $Q_0$ given by

$$Q_0 = \int\limits_B M_0 \phi^n k_{zz} \left( -\frac{\partial p^l}{\partial z} + \Delta \rho g \right) \mathrm{d}x. \tag{14}$$

This is an integral of the vertical component of Eq. (10) over $B$. Assuming a constant pressure gradient ($\partial p^l / \partial z$) in the region where $\phi > 0$ at the bottom boundary, we can rearrange Eq. (14) as a boundary condition for $\partial p^l / \partial z$.

As we demonstrate below, this combination of domain, boundary and initial conditions are appropriate choices to simulate

the poro-VEVP equivalent of dykes. We analyse their behaviour with reference to the poro-LEFM dyke model.

It should be noted that we do not prescribe the pressure gradient at the bottom boundary of the poro-VEVP model as $\partial p^l / \partial z = 0$, which is the far-field condition of the poro-LEFM model. This is primarily due to the limitation inherent in the finite computational domain and further affected by the two-dimensonality in the poro-VEVP model. Firstly, a finite domain cannot simulate an infinitely long dyke, thus the bottom boundary cannot be treated with a far-field condition. Secondly,

unlike the poro-LEFM model which only considers horizontal displacement, the 2D continuum model allows for both vertical and horizontal deformation within the dyke due to solid phase (de)compaction. This results in a more complex solid stress tensor that must be balanced by the liquid pressure. These solid stresses remain significant even further away from the dyke tip, contrasting with the zero elastic solid pressure assumed in the poro-LEFM model (details in Appendix F). Given these limitations, we define a constant liquid volume rate $Q_0$ instead. The propagation rate of the tip is a key point of comparison

with the poro-LEFM model. To quantify it, we define a tip location $z_t$ as the highest point along the vertical cross section at $x = 0$ km where $\phi \geq 10^{-3}$. The tip speed is then diagnosed from the numerical results as $v_t = \mathrm{d}z_t/\mathrm{d}t$.

### 2.3   Energy analysis and the effective toughness

This section analyses the energy budget of the poro-VEVP model of a dyke. It estimates the effective fracture toughness in terms of the rate at which mechanical energy is dissipated by the propagation of the dyke tip.

In the poro-VEVP model, the total work rate $\dot{W}$ deforming the solid phase over a domain $\Omega$ is written as

$$\dot{W} = \int\limits_\Omega \dot{w} \, \mathrm{d}A, \quad \text{with} \quad \dot{w} = \dot{w}^{\mathrm{v}} + \dot{w}^{\mathrm{e}} + \dot{w}^{\mathrm{K}}. \tag{15}$$



Here, $\dot{w}$ is the local work rate at a point, decomposed into viscous $\dot{w}^{\mathrm{v}}$, elastic $\dot{w}^{\mathrm{e}}$, and viscoplastic $\dot{w}^{\mathrm{K}}$ components for this Maxwell material. Appendix D provides details of the formulation for each local work rate. The total poro-VEVP work rate is similarly decomposed as

$$\dot{W} = \dot{W}^{\mathrm{v}} + \dot{W}^{\mathrm{e}} + \dot{W}^{\mathrm{K}}. \tag{16}$$


This can be compared with the (poro-)LEFM model, where the work rate includes elastic and fracture components,

$$\dot{W}_{\mathrm{LEFM}} = \dot{W}^{\mathrm{e}}_{\mathrm{LEFM}} + \dot{W}^{\mathrm{f}}_{\mathrm{LEFM}}, \tag{17}$$

where the term with superscript f is the work rate to create new surface area of the fracture.

As a basis for comparison of a steadily propagating, constant flux, poro-VEVP dyke with a poro-LEFM dyke under the same
conditions, we require that $\dot{W} = \dot{W}_{\mathrm{LEFM}}$. Then, assuming that the elastic contributions to these work rates are approximately equal, we obtain a relationship between the dissipative parts,

$$\dot{W}^{\mathrm{v}} + \dot{W}^{\mathrm{K}} \approx \dot{W}^{\mathrm{f}}_{\mathrm{LEFM}}. \tag{18}$$

We can use this result to diagnose a fracture toughness for the LEFM model.

In LEFM theory, the energy expended to propagate the fracture a unit distance is the fracture toughness $\mathcal{G}$ (Anderson, 2017).
We adopt the same definition in the poro-LEFM model and assume a constant propagation speed $c = v_t$, i.e., an identical speed between the two formulations. Thus the fracture energy rate is $\dot{W}^{\mathrm{f}}_{\mathrm{LEFM}} = \mathcal{G}v_t$ which is the fracture energy budget per unit time. Combining this with equation (18), we calculate fracture toughness $\mathcal{G}$ and critical stress intensity $K_c$ from the dissipation rate of the poro-VEVP model as

$$\mathcal{G} = \frac{\dot{W}^{\mathrm{v}} + \dot{W}^{\mathrm{K}}}{v_t}, \qquad K_c = \left(\frac{2G\mathcal{G}}{1-\nu}\right)^{1/2} = \left[\frac{2G(\dot{W}^{\mathrm{v}} + \dot{W}^{\mathrm{K}})}{v_t(1-\nu)}\right]^{1/2}. \tag{19}$$

The second equation is obtained from the LEFM relationship between the critical stress intensity factor and the fracture toughness for plane-strain deformation (Anderson, 2017), with substitution of the first equation for the fracture toughness in terms of the poro-VEVP dissipation rate.

As noted above, we suppress viscous deformation by prescribing a Maxwell viscosity that is effectively infinite (without changing the problem formulation). Because of this we have $\dot{W}^{\mathrm{v}} \approx 0$ and hence the dissipation in the poro-VEVP model is
entirely viscoplastic. Furthermore, we choose a small viscoplastic viscosity $\eta^{\mathrm{K}}$ to reduce the viscous dissipation in the Kelvin component. Appendix D2 discusses the effect of $\eta^{\mathrm{K}}$ on the rate of mechanical energy dissipation.





**Table 1.** Dimensional parameters for computational modelling.

| Parameter | Name | Unit | Value |
|---|---|---|---|
| $\rho^s$ | Solid density | kg m$^{-3}$ | 3000 |
| $\rho^l$ | Melt density | kg m$^{-3}$ | 2500 |
| $\eta^K$ | Viscoplastic viscosity | Pa s | $10^{10}$ |
| $G$ | Reference shear modulus | GPa | 5 |
| $Z$ | Reference bulk modulus | GPa | 10 |
| $C$ | Cohesion | MPa | 5 |
| $\theta$ | Friction angle | ° | 30 |
| $\sigma_t$ | Tensile strength | MPa | 1.25 ($\sigma_t = C/4$) |
| $M_0$ | Mobility prefactor | m$^2$(Pa s)$^{-1}$ | $10^{-9}$ |
| $n$ | Exponent in the permeability-porosity relation | - | 3 |
| $g$ | Gravity constant | m s$^{-2}$ | 9.8 |
| $\phi_{bg}$ | Background porosity | | $10^{-10}$ |
| $Q_0$ | Liquid volume flux rate | m$^2$ yr$^{-1}$ | 40 |
| $k_a$ | Maximum permeability enhancement | - | 10 |
| $\epsilon$ | Characteristic anisotropy of plastic strain | - | 0.05 |

## 3 Results

The results are divided into two parts. First we document the output of the poro-VEVP model in terms of its dyke-like solutions.
Second, we describe the comparison of those solutions to the poro-LEFM model.

### 275 3.1 Results of the poro-VEVP model

This section presents numerical solutions of the poro-VEVP model. We first analyse a reference case (parameters listed in
Table 1) that demonstrates a steadily propagating dyke. We then investigate the effects of varying viscoplastic viscosity ($\eta^K$)
and maximum permeability enhancement ($k_a$).

Figure 3(a) shows a snapshot of the porosity field from a representative numerical solution. The field includes a porous dyke
with uniform width that rises up through the middle of the domain. A close-up investigation reveals that the central column
of cells holds $> 90\%$ of the total volume of liquid in the domain. The porosity in laterally adjacent cells is at least ten times
smaller. This shows a negligible leakage through the wall and can confine the porous dyke to one cell in width. This width
remains constant over time, enabling one-dimensional analysis along the central column of cells that represents the dyke. While
advantageous in terms of comparison with a poro-LEFM model in which dykes are narrow relative to our grid spacing, this
pattern raises questions about the grid-size dependence of the results. Because this dyke has a width of $\Delta x$, holding the volume
rate $Q_0$ constant implies that the volume flux ($Q_0/\Delta x$) varies with the grid spacing. Consequently, the boundary condition
for the pressure gradient in Eq. (14) also changes, significantly impacting the results. However, if we instead fix the pressure
boundary condition and therefore the volume flux, we find the grid size has little impact; the relative variation in porosity is
$< 5\%$ when $\Delta x$ is further reduced from 40 m to 30 m.





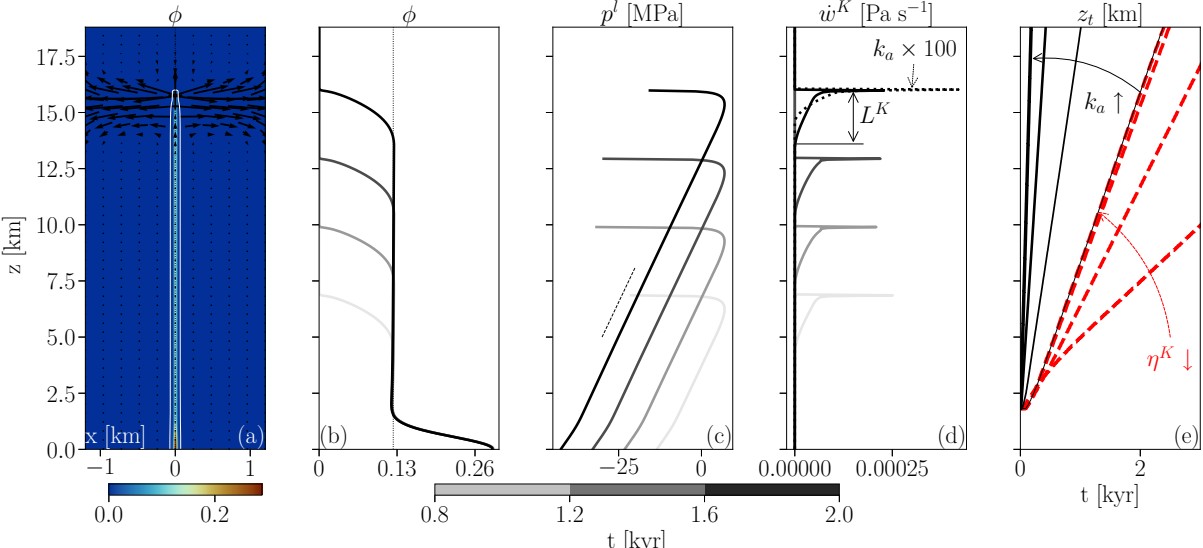

**Figure 3.** Results from a reference calculation of the poro-VEVP model. (a) Porosity and solid deformation field at $t = 2$ kyr. The white curve represents the contour of $\phi = 10^{-3}$. (b) Profiles of $\phi$ (solid lines) along a vertical cross section at $x = 0$ for $t = 0.8$, 1.2, 1.6, and 2.0 kyr. The dotted line represents $\phi = 0.13$. (c) Profiles of $p^l$ along $x = 0$ at the same time-steps as (b). The slope in the tail region matches the poro-LEFM prediction with a prescribed flow rate and porosity (the dotted line). (d) The corresponding local plastic dissipation rate along $x = 0$. Solid lines represent the reference case with $k_a = 10$; the dotted line shows a case with $k_a = 10^3$ for comparison. The region below the tip with non-zero $\dot{w}^K$ is referred to as the head region; its size is denoted $L^K$. (e) Tip propagation for different $\eta^K$ (red) and $k_a$ (black). Dashed red lines show propagation rate convergence for decreasing $\eta^K$ ($10^{18}$, $10^{17}$, $10^{16}$, and $10^{15}$ Pa s), as indicated by red arrows. The last one converges to the reference case, $10^{10}$ Pa s (thin solid line), with a speed of $v_t = 7.6$ m/yr, matching the poro-LEFM prediction. The black lines show the variation of propagation speed for $k_a = 10$ (reference case), $10^2$, $10^3$, and $10^4$.

Figure 3(b)–(d) illustrates the steady advance of the dyke tip and the liquid phase. Panel (b) depicts porosity $\phi$; panel (c) depicts liquid pressure $p^l$; panel (d) depicts local plastic dissipation rate $\dot{w}^K$. Each panel shows four curves at different times (0.8, 1.2, 1.6, and 2.0 kyr), confirming that the tip advances approximately the same distance in each 0.4 kyr interval. In panel (b) there is a region at $z < 2$ km where the interior solution adjusts to match the boundary condition. Above this, for all four times, there is a region with uniform $\phi \approx 0.13$. The height of this region grows linearly with time. Above this uniform

region, each curve has a region where the porosity varies from $\phi \approx 0.13$ to zero at the dyke tip.

Panel (d) shows that beneath the tip is a region where plastic work is done. Indeed the position of the tip is characterised by the spike in $\dot{w}^K$. We define the head of the poro-LEFM dyke as where $\dot{w}^K$ is non-zero—that is, the entire region experiencing plastic tensile failure. In the reference case, this region is about 2.4 km high and confined to the column of grid cells that contain liquid. This height reduces to about 1.3 km when the permeability enhancement is 100 times larger (dotted line). The

head region has a prominent solid displacement rate as shown in panel (a). At the dyke tip, panel (c) shows that the pressure gradient is nearly singular; this is the location of tensile yielding, also corresponding with the spike in dissipation rate.

The mechanics of the head region represents a key difference between the poro-VEVP and poro-LEFM models. In the poro-VEVP model, buoyancy induces plastic tensile failure throughout the head region, whereas in the poro-LEFM model, fracture





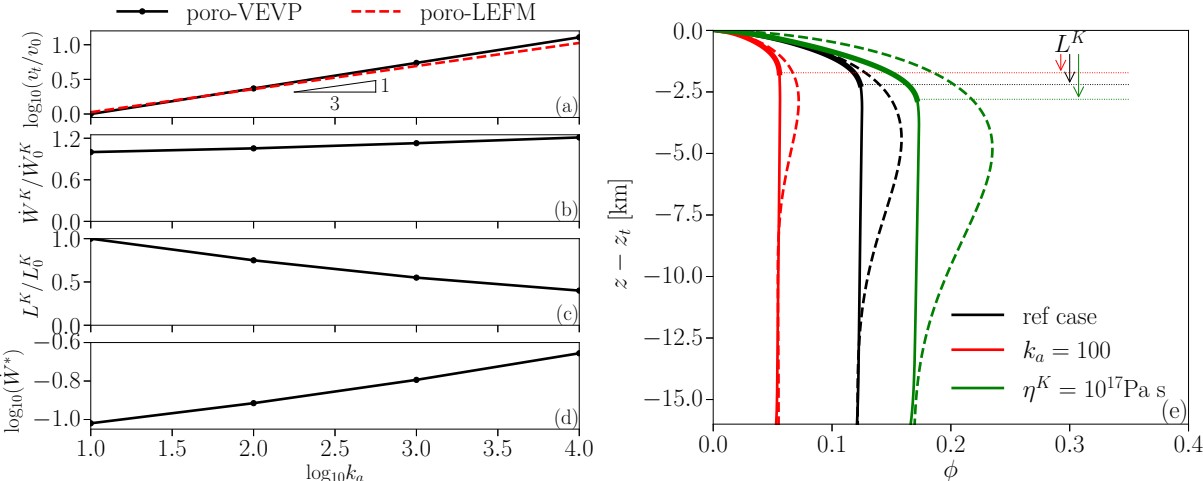

**Figure 4.** Key characteristics of simulated dykes as a function of the log of $k_a$. (a) The power-law relation between tip speed $v_t$ and anisotropic permeability enhancement $k_a$. Here $v_t$ is scaled by the speed in the reference case, $v_0 = 7.6$ m/yr. (b) The rate of plastic work $\dot{W}^{\mathrm{K}}$ increases with $k_a$, but only by $\sim 7\%$ or less for each tenfold increase of $k_a$. Here $\dot{W}^{\mathrm{K}}$ is scaled by the reference result $\dot{W}_0^{\mathrm{K}} = 3.1$ Pa/s. (c) The size of plastic zone $L^{\mathrm{K}}$ decreases with $k_a$. The scaling factor is the reference result $L_0^{\mathrm{K}} = 2.4$ km. (d) The dissipation intensity at the tip $\dot{W}^*$ increases with $k_a$. It measures the ratio of the dissipation rate in the tip cell to the overall rate. (e) Comparison of the porosity profiles of the poro-VEVP model (solid lines) with the poro-LEFM model (dashed lines). The critical stress intensities are $K_c = 0.51$ (black), 0.34 (red), and 1.08 (green) GPa m$^{1/2}$, calculated using the energy analysis of the poro-VEVP model (Eq. (19)). Thicker solid lines indicate plastic zones in each case.

is localised exclusively to the tip. This difference is reflected in the pattern of energy dissipation of each model: distributed

over a finite zone in poro-VEVP versus localised to a point in poro-LEFM.

The tail region in Fig. 3(c) shows another distinction between the two models. The poro-VEVP model has a constant, non-zero pressure gradient $\partial p^l/\partial z \approx 3.2$ MPa/km in the tail, contrasting to the zero far-field pressure gradient in the poro-LEFM model. This distinction stems from the limitations of the finite domain and the significant solid stress gradient, which necessitates a balancing liquid pressure gradient. This prevents the use of a zero pressure gradient as a boundary condition on

the bottom, as explained in Section 2.2.4 and further detailed in Appendix F.

Figure 3(e) shows tip propagation at various values of viscoplastic viscosity $\eta^{\mathrm{K}}$ and permeability anisotropic enhancement $k_a$. All curves become linear in time after a short transient, indicating constant propagation speed. Speed increases as $\eta^{\mathrm{K}}$ decreases from $10^{18}$ Pa s, but converges to a constant value below $10^{15}$ Pa s. Increasing $k_a$ further increases the tip speed.

Figure 4(a) confirms the effect of $k_a$ on $v_t$ in a log–log plot, indicating a power-law relationship arising from the mobility

closure,

$$v_t \propto k_a^{1/3}, \tag{20}$$





at constant influx rate. This relationship informs the choice of time-step size to ensure the accuracy by maintaining a moderate Courant number.

Panel (a) also demonstrates the agreement between the power-law relationship measured in poro-VEVP numerical solutions and the analytical prediction of the poro-LEFM model. Use of Darcy's law in the poro-LEFM model requires that $Q_0 \propto M_\phi$ for constant width, which translates to $Q_0 \propto k_a \phi_0^n$ when fluid mobility is matched to the poro-VEVP model. This relationship indicates that, for a fixed $Q_0$, varying the permeability enhancement adjusts the far-field porosity according to $\phi_0 \propto k_a^{-1/n}$. Given that $Q_0 = \phi_0 h c$ in the far field, the propagation speed must therefore scale as $c \propto k_a^{1/n}$. Recalling that we choose $n = 3$, this scaling governs propagation speed in both models, despite their different values resulting from distinct pressure gradients in the tail region.

Figure 4(b) shows that the overall plastic dissipation rate increases with permeability enhancement, but only by 20% over a factor of $10^3$ change in $k_a$. This change is negligible compared to the tenfold increase in propagation speed shown in panel (a). Therefore we can consider the total dissipation rate to be essentially independent of $k_a$. Recalling the calculation for fracture toughness and critical stress intensity in Eq. (19), we obtain the following power-law relationship for $\mathcal{G}$ and $K_c$ in terms of $k_a$,

$$\mathcal{G} \propto k_a^{-1/3}, \quad K_c \propto k_a^{-1/6}. \tag{21}$$

This contrasts with (poro-)LEFM models, where fracture toughness is independent of permeability, while the fracture energy rate changes in proportion to propagation speed.

Figures 4(c) and (d) show that larger permeability enhancement leads to a shorter plastic zone $L^{\mathrm{K}}$, meaning a smaller head region and more intense plastic dissipation at the tip. This intensity is measured by the ratio of dissipation rate in the tip cell to the overall dissipation rate, $\dot{W}^* \equiv \dot{w}_m^{\mathrm{K}} \Delta x \Delta z / \dot{W}^{\mathrm{K}}$. Here $\dot{w}_m^{\mathrm{K}}$ denotes the work rate at the tip, which corresponds to the maximum value of curves in Fig. 3(d). Given that $\dot{W}^{\mathrm{K}}$ is constant when fixing $Q_0$ and varying $k_a$, Fig. 4(d) also represents the variation of the peak dissipation rate $\dot{w}_m^{\mathrm{K}}$ as a function of the permeability enhancement. Together, panels (c) and (d) indicate that increasing $k_a$ reduces head height $L^K$ and focuses plastic failure onto the tip. This trend provides an explanation for the reduction of fracture toughness associated with increasing $k_a$.

## 3.2 Comparison between the poro-VEVP and poro-LEFM models

This section compares the poro-VEVP and poro-LEFM dykes in terms of porosity profiles and stress distribution. We impose that the poro-LEFM dyke has the same width as the poro-VEVP dyke and has a far-field porosity equal to the tail-region porosity. Based on the energy analysis of the poro-VEVP results, we estimate an effective fracture toughness $\mathcal{G}$ and thus a critical stress-intensity factor $K_c$, which we then apply to the poro-LEFM model. In the comparison below, we evaluate whether this estimated $K_c$ is an appropriate value to link these two models.

On the basis of this estimated $K_c$, Figure 4(e) compares porosity profiles between the poro-VEVP (solid lines) and poro-LEFM (dashed lines) models. The panel shows three cases: the reference case (black), a case with increased viscoplastic viscosity $\eta^{\mathrm{K}}$ (green), and a case with increased maximum permeability enhancement $k_a$ (red). When $\eta^{\mathrm{K}}$ is relatively small



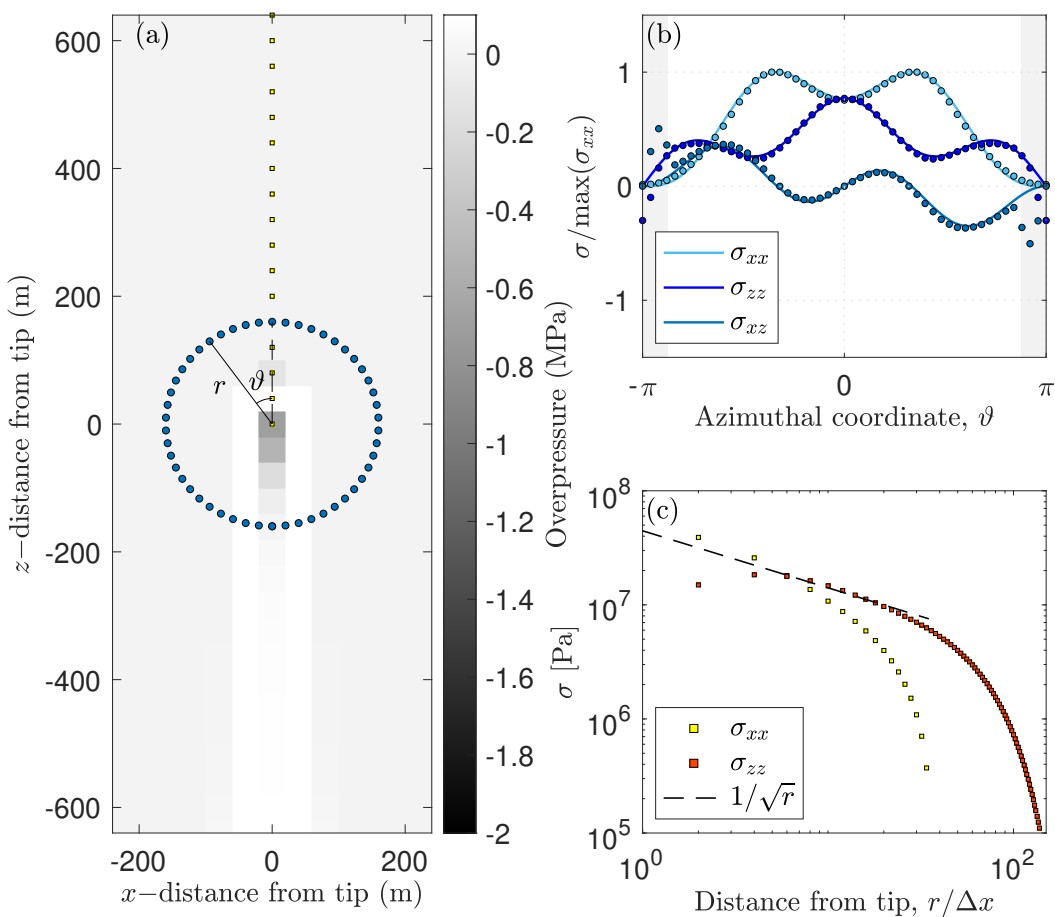

**Figure 5.** Comparison of stress components between the poro-VEVP model and the (poro-)LEFM model with $K_c = 510$ MPa m$^{1/2}$. (a) Fracture-tip coordinate system, where angle $\vartheta$ is measured counter-clockwise from the vertical axis and radial distance $r$ is measured from the origin. The background grey scale represents the liquid overpressure $\Delta P$. (b) Fracture-tip stress distribution. LEFM solutions are depicted as solid lines, while poro-VEVP stress components as points evenly spaced in $\vartheta$ around the fracture at $r = 160$ m (indicated by blue dots in panel (a)). Regions where $|\vartheta| > 7\pi/8$ are shaded grey. (c) Fracture-tip stress asymptote. Squares represent poro-VEVP results directly ahead of the fracture tip (along $\vartheta = 0$; yellow points in panel (a)). The dashed line represents the LEFM $1/\sqrt{r}$ singularity.





(black and red lines), the continuum and fracture models match well near the tip, suggesting that the plasticity-based $K_c$ (and thus $\mathcal{G}$) can quantitatively relate these two models. However, when $\eta^{\mathrm{K}}$ is relatively large (green), the two models are not closely aligned, even near the tip. Considering all three cases, we notice that the poro-VEVP dykes do not have the bulbous head which appears in the poro-LEFM dykes. What we have defined as the head in the poro-VEVP model ($L^{\mathrm{K}}$), where plastic failure takes place, is much shorter than the head height in the poro-LEFM model.

Figure 5 compares components of the stress tensor between the two models in the zero-porosity region. The tensor is evaluated at points (blue dots in panel (a)) around a circle centred at the dyke tip, and along a vertical line upwards from the tip (yellow dots in panel (a)). The stress calculation for the poro-LEFM is presented in Appendix E, which is identical to the LEFM model in the zero-porosity region. Panel (b) shows agreement of stress components between poro-VEVP and (poro-)LEFM along the azimuthal coordinate $\vartheta$ along a circle of radius $r = 160$ m ($= 4\Delta x$) in the region $\vartheta \in [-7\pi/8, 7\pi/8]$. Regarding the stress distribution along the radial direction, the (poro-)LEFM model predicts that $\sigma_{xx}$ and $\sigma_{zz}$ are both proportional to $1/\sqrt{r}$, where $r$ is distance from the tip. Panel (c) shows that the poro-VEVP results is somewhat but not entirely consistent with this prediction; $\sigma_{xx} \sim r^{-1/2}$ when $r < 5\Delta x$ and $\sigma_{zz} \sim r^{-1/2}$ when $r \in [4, 16]\Delta x$.

Li et al. (2023) made a similar comparison of the stress distribution between models, but for the case of a dyke driven by uniform horizontal tension, imposed in the far field. The present manuscript enhances the credibility of such a comparison in two key ways: first, the poro-VEVP dyke is driven purely by buoyancy, consistent with the (poro-)LEFM dyke; second, the stress intensity factor is derived from the plastic dissipation rate of the poro-VEVP model, rather than using a fitted value.

## 4 Discussion

In the preceding sections, we compared poro-VEVP and LEFM models for simulating buoyancy-driven dykes. The comparison was facilitated by the introduction of an intermediary poro-LEFM model. This section discusses the results and addresses the slow propagation and high toughness of poro-VEVP dyking.

This study demonstrates that the poro-VEVP model can represent dykes with plastic tensile failure. Specifically, by incorporating anisotropic permeability, this model can simulate a long, thin dyke-like melt conduit with minimal liquid leakage through the walls, such that it is generally consistent with an LEFM model. The dyke width is determined by the grid size, which is a limitation of the present discretized solutions of the continuum models. Despite this limitation, we can validate the poro-VEVP model against a poro-LEFM model, comparing the porosity and stress distributions of dykes with the same width.

The slow propagation speed of poro-VEVP dykes arises from the large drag on fluid motion under Darcy flow compared to Poiseuille flow in the LEFM model. This is quantified by the mobility $M_\phi$, the ratio of permeability to liquid viscosity. Mobility is parameterised in terms of the product of a prefactor $M_0$ and a power of the porosity $\phi$. While $\phi$ is part of the solution and cannot be directly manipulated to control the speed, $M_0$ can be increased within a dyke by prescribing a permeability enhancement $k_a$. Above we showed that the speed increases with $k_a$ following the power law, $\propto k_a^{1/n}$, when the liquid volume influx is fixed. However, a faster dyke requires a smaller time-step for accuracy, thereby increasing the computational cost.





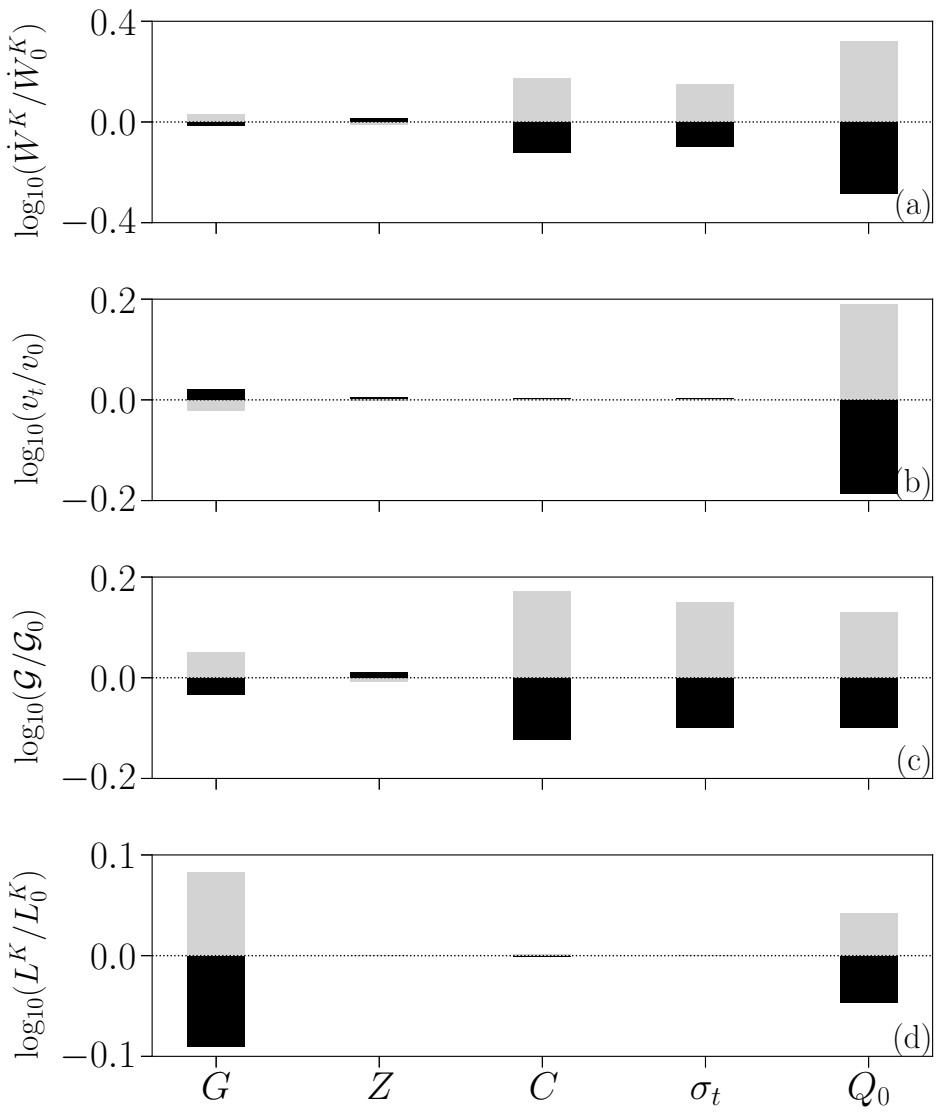

**Figure 6.** Dependence of the overall dissipation rate $\dot{W}^{\mathrm{K}}$, propagation speed $v_t$, fracture toughness $\mathcal{G}$, and the head height $L^{\mathrm{K}}$ on the physical parameters: the shear modulus $G$, the bulk modulus $Z$, the cohesion $C$, the tensile strength $\sigma_t$, and the volume influx rate $Q_0$. Gray and black bars in each panel represent the variation of each variable by changing $\times 2$ and $\times 1/2$, respectively, on the reference value of each parameter. The variation is shown as the change relative to the corresponding result in the reference case.





Therefore, when using the poro-VEVP model, consideration must be given to balancing the desire for more accurately rapid dyke propagation with the computational cost this incurs.

The fracture toughness of poro-VEVP dykes can be calculated from the plastic dissipation energy of the continuum model by assuming its equivalence to the fracture energy in the poro-LEFM model. In this way, we relate the toughness value to the
speed of tip propagation and the size and intensity of the distributed plastic failure over a head region close to the dyke tip. This region is much shorter than the bulbous head in the poro-LEFM model, defined by where the porosity is distinct from the far-field porosity (2.4 km versus 12 km for the reference case). The finite-size failure region represents another difference to the poro-LEFM model, in which fracture occurs at the tip only. Despite this, by using the estimated toughness in the poro-LEFM model, we achieve reasonable agreement in the porosity profiles and stress distribution between the two models.

This toughness value is influenced by various physical parameters that alter the dynamics in the head region, including permeability enhancement ($k_a$), shear ($G$) and bulk ($Z$) modulus, cohesion ($C$), tensile strength ($\sigma_t$), and volume flux rate ($Q_0$). Figure 4 shows that increasing $k_a$ leads to a decrease in $\mathcal{G}$, while Figure 6 demonstrates a positive correlation between $\mathcal{G}$ and increasing values of $G$, $C$, $\sigma_t$, and $Q_0$. The elastic bulk modulus $Z$ does not have a significant effect on $\mathcal{G}$.

These parameters affect fracture toughness in different ways. Increasing $k_a$ reduces the head height and localises dissipation
to the tip (Fig. 4), resulting in a reduced toughness. A similar relationship between the localisation of plastic dissipation and toughness is obtained by varying the elastic shear modulus (Fig. 6, first column). Increasing $G$ leads to increased toughness, accompanied by a longer plastic zone with a similar total dissipation rate, meaning a more distributed failure and taller head. Increasing cohesion and tensile strength also increases toughness, but it does so by increasing the overall dissipation rate without affecting the size of the plastic zone (Fig. 6, third and fourth column). In these cases, the strength of plastic failure,
rather than its distribution, is the primary factor associated with the variation of toughness. Furthermore, while a higher liquid volume flux increases the overall dissipation rate more than cohesion or tensile strength, it has a lesser effect on fracture toughness (Fig. 6, fifth column). This can be attributed to the increased propagation speed, which lowers the dissipation work per unit length of fracture growth.

The dependence of toughness on liquid volume flux is intriguing because, in the poro-LEFM model, liquid-phase dynamics
do not affect solid properties. This may be explained in terms of two related ideas. First, the toughness as evaluated in poro-VEVP is associated with the energetics of the head region. This region has non-zero porosity, making the dissipation a property of the two-phase medium, i.e., something affected by the liquid phase. In contrast, the non-zero porosity in the poro-LEFM dyke does not affect the fracture energy because the fracture occurs precisely at the tip, where the porosity is zero.

Second, this sensitivity of toughness to liquid flux resembles that of more complex fracture mechanics theories like Elastic
Plastic Fracture Mechanics (EPFM) (Anderson, 2017). EPFM applies a plastic yield limit to an elastic fracture-mechanics model. On this basis it predicts a plastic zone around the fracture tip, where the intensified elastic stress reaches the yield limit. Papanastasiou (1999) uses EPFM to model a constant-flux fluid-driven fracture, showing that a higher liquid flux leads to a larger plastic zone and, consequently, higher effective toughness and stress intensity. A large toughness and stress intensity in the poro-VEVP model can therefore be broadly related to plastic dissipation in the EPFM model. In fact, observations suggest
that a large toughness might be possible in the field: Gudmundsson (2009) suggests a toughness value in volcanic edifices two



orders of magnitude larger than that reported by laboratory experiments. To quantitatively align the poro-VEVP model with both EPFM model and field observations is beyond the present scope.

One limitation of the present research arises from the simplified form of anisotropic permeability that we impose. In particular, our formulation modifies only the horizontal or vertical permeability. This is appropriate if the dyke (or sill) aligns with one of these two directions, but it is unsuitable for modelling curved dyke trajectories, such as those influenced by ambient stresses (Maccaferri et al., 2014). Thus, the formulation of anisotropic permeability needs to be generalised to enable dyke propagation in an arbitrary direction. We will address this in future work.

Another limitation of this work is the difference in boundary conditions between poro-VEVP (constant volume flux, leading to a non-zero pressure gradient) and poro-LEFM (zero pressure gradient at the far-field). This is, however, unavoidable because of the limitations of the finite domain and also the two-dimensionality of the continuum model. As a result, the stresses between the solid and liquid balance differently inside of the dyke (see Appendix F for details). Nonetheless, we still achieve a reasonable agreement between the two models near the tip by assuming the equivalence between plastic dissipation and fracture energy.

In conclusion, with some caveats, the representation of a dyke in the continuum, poro-VEVP formulation is consistent with linear elastic fracture mechanics. This consistency supports the validity of our approach for geodynamic applications. Moreover, it gives us confidence in incorporating poro-VEVP into large-scale rifting models requiring consistent magma transport in both ductile and brittle regions of the lithosphere (e.g. Pusok et al., 2024).

## 5 Summary

This study compares dyke propagation in a poro-viscoelastic–viscoplastic model with that in a canonical linear elastic fracture mechanics model. The comparison is enabled by interposing a novel poro-LEFM model. It highlights two key discrepancies: slow propagation speed of the poro-VEVP dyke, and the requirement for large fracture toughness in the LEFM model to match the poro-VEVP results. We have reported on our progress in addressing these discrepancies.

Slow propagation speed in the poro-VEVP model is primarily attributed to low permeability relative to an open fracture. This limitation can be mitigated by introducing an anisotropic permeability enhancement. The large equivalent toughness value inferred for the poro-VEVP model can be explained in terms of plastic dissipation of mechanical energy. This effective fracture toughness depends on various physical parameters that affect the plastic dissipation rate in the solid–liquid aggregate. The poro-VEVP models now incorporates a new formulation for the constitutive relation between compaction stress and strain rates, which improves solver reliability over that used by (Li et al., 2023). Future development will focus on implementing the full anisotropic permeability tensor to investigate how the ambient stress field influences dyke (or sill) emplacement.

*Code availability.* The current version of model is available at https://github.com/YuanLiAC/poroVEVP under the MIT licence. The exact version of the model used to produce the results used in this paper is archived on Zenodo (Li et al., 2024), as are input data and scripts to



run the model and produce the plots for all the simulations presented in this paper. The poro-VEVP model has dependencies on FD-PDE (Pusok et al., 2022a) and PETSc (Balay et al., 2022b). Visualisation and post-processing utilised the colour scheme from Scientific Color Maps (Crameri et al., 2020; Crameri, 2021). Full simulation data can be provided by YL on request.

## Appendix A: Mathematical formulation of the poro-LEFM model

This section provides details of the mathematical formulation of the poro-LEFM model that was introduced in section 2.1. It explains how the governing equations of the liquid and solid phases are obtained and how they are non-dimensionalised.

### A1  The liquid phase

We derive a mass continuity equation for the poro-LEFM model from Darcy's law and the mass conservation equation of a two-phase continuum model,

$$\phi(\boldsymbol{v}^l - \boldsymbol{v}^s) = -M_\phi \left( \boldsymbol{\nabla} P^l - \rho^l \boldsymbol{g} \right), \tag{A1}$$

$$\frac{\partial \phi}{\partial t} + \boldsymbol{\nabla} \cdot \left( \phi \boldsymbol{v}^l \right) = 0. \tag{A2}$$

We decompose the full liquid pressure gradient, $\boldsymbol{\nabla} P^l$, into static and dynamic components as

$$\boldsymbol{\nabla} P^l = \rho^s \boldsymbol{g} + \boldsymbol{\nabla} p^l. \tag{A3}$$

We denote the vertical component of liquid and solid velocity by $v^l$ and $v^s$. Taking the vertical component of Eq. (A1) and assuming zero vertical solid velocity ($v^s = 0$), we obtain the liquid flux rate as

$$\phi v^l = M_\phi \left( -\frac{\partial p^l}{\partial z} + \Delta \rho g \right). \tag{A4}$$

We assume zero horizontal component of liquid velocity, which implies no leakage through the fracture wall. Then Eq. (A2) reduces to

$$\frac{\partial \phi}{\partial t} + \frac{\partial \phi v^l}{\partial z} = 0. \tag{A5}$$

For an infinitely long buoyancy-driven dyke, we expect uniform propagation at a fixed speed with constant far-field porosity. Assuming pure buoyancy drive (i.e., $\partial p^l / \partial z = 0$ at the far field), Eq. (A4) yields the constant propagation speed,

$$c = \frac{M_\phi(\phi_0)}{\phi_0} \Delta \rho g, \tag{A6}$$

where $\phi_0$ is the far-field porosity. In this case, the far-field liquid volume rate is $Q_0 = \phi_0 h c$.



## A2 Solid and liquid stresses

We formulate the elastic solid stress distribution $p^s(z,t)$ of the poro-LEFM model following the LEFM model (e.g., Weertman, 1971; Lister, 1990; Roper and Lister, 2007). This elastic stress, associated with dyke opening in the $x$ direction, intensifies towards infinity at the tip, characterised by a critical stress intensity $K_c$.

The mathematical formulations are

$$p^s(z,t) = -\left(\frac{G}{1-\nu}\right)\frac{1}{2\pi}\int_{-\infty}^{\infty}\frac{\partial h\phi(\xi,t)}{\partial\xi}\frac{\mathrm{d}\xi}{\xi-z}, \tag{A7}$$

$$p^s(z,t) = -\frac{K_c}{[2(z-z_t)]^{1/2}}, \quad \text{at } z \to z_t^+. \tag{A8}$$

Here, $h\phi$ represents the horizontal deformation required to open a porous dyke of width $h$ with porosity $\phi$, and $z_t$ is the tip location.

We assume force balance between the solid and liquid phases in the non-zero porosity region, so $p^l = p^s$ across the dyke in the poro-LEFM model.

## A3 Non-dimensionalisation

We transform the coordinate system to be fixed with respect to the fracture tip, changing $(z,t)$ to $z'(t) = (z_t^o + ct) - z$, where $z_t^o$ is the initial tip location.

We take the following non-dimensionalisation,

$$\tilde{\phi} = \frac{\phi}{\phi_0}, \quad \tilde{z} = z'\left(\frac{Gh\phi_0}{(1-\nu)\Delta\rho g}\right)^{-1/2}, \quad \tilde{p}^s = p^s\left(\frac{Gh\phi_0}{(1-\nu)}\Delta\rho g\right)^{-1/2}, \quad \tilde{K}_c = K_c\left(\frac{Gh\phi_0}{1-\nu}\right)^{-3/4}(\Delta\rho g)^{-1/4}. \tag{A9}$$

The system of governing equations (Eqs. (1) – (4)) then leads to the following non-dimensionalised system,

$$\frac{\mathrm{d}\tilde{p}^s}{\mathrm{d}\tilde{z}} = \left(\frac{1}{\tilde{\phi}}\right)^2 - 1, \tag{A10}$$

$$\tilde{p}^s(\tilde{z}) = -\frac{1}{\pi}\int_{0}^{\infty}\frac{\mathrm{d}\tilde{\phi}(\xi)}{\mathrm{d}\xi}\frac{\mathrm{d}\xi}{\xi-\tilde{z}}, \tag{A11}$$

$$\tilde{p}^s \approx -\frac{\tilde{K}_c}{(-2\tilde{z})^{1/2}}, \text{ at } \tilde{z} \to 0^-, \tag{A12}$$

$$\tilde{\phi} \approx 1, \text{ at } \tilde{z} \to \infty. \tag{A13}$$



## Appendix B: A new formulation of $\Delta P$ in the poro-VEVP model

This section addresses an issue with representing the constitutive law for $\Delta P$ using the effective viscosity approach and presents a new formulation to resolve this issue. This constitutive law relates the compaction stress $\Delta P$ to the compaction rate $\mathcal{C}$.

We recall that the compaction rate for a poro-VEVP rheology as

$$\mathcal{C} = \mathcal{C}^{\mathrm{v}} + \mathcal{C}^{\mathrm{e}} + \mathcal{C}^{\mathrm{K}}, \tag{B1}$$

where superscripts v, e, and K represent viscous, elastic, and viscoplastic components, respectively. Substituting the rheological models of the viscous and elastic components into the right-hand side of (B1) (cf. Li et al. (2023)), we rearrange the resulting formulation as

$$(1-\phi)\Delta P = -\zeta^{\mathrm{ve}}(\mathcal{C}' - \mathcal{C}^{\mathrm{K}}), \quad \text{where } \zeta^{\mathrm{ve}} = \left(\frac{1}{\zeta_\phi^{\mathrm{v}}} + \frac{1}{Z_\phi \Delta t}\right)^{-1}, \quad \mathcal{C}' = \left[\mathcal{C} - \frac{(1-\phi)\Delta P^o}{Z_\phi \Delta t}\right]. \tag{B2}$$

Here, $\zeta_\phi^{\mathrm{v}}$ and $Z_\phi$ are the compaction viscosity and bulk modulus, respectively, $\Delta t$ is the time-step size, $\Delta P^o$ is the overpressure at the previous time-step, and $\mathcal{C}^{\mathrm{K}}$ is the plastic compaction rate.

The effective viscosity approach assumes

$$(1-\phi)\Delta P = -\zeta_{\mathrm{eff}}\mathcal{C}', \tag{B3}$$

where $\zeta_{\mathrm{eff}}$ is held constant when solving the force-balance equation for strain rates. It is determined as follows. If there is no plastic yielding or no dilatancy when yielding (i.e., $\mathcal{C}^{\mathrm{K}} = 0$), then $\zeta_{\mathrm{eff}} = \zeta^{\mathrm{ve}}$. Otherwise, when $\mathcal{C}^{\mathrm{K}} \neq 0$, $\zeta_{\mathrm{eff}} = -\frac{(1-\phi)\Delta P}{\mathcal{C}'}$, where $\Delta P$ is calculated using the return mapping method (Krieg and Krieg, 1977) to constrain stresses on the yield surface. However, $\zeta_{\mathrm{eff}}$ becomes infinite when $\mathcal{C}' = 0$ and $\Delta P \neq 0$. In this circumstance, the effective viscosity approach is no longer appropriate.

To address this issue, we propose an alternative formulation of $\Delta P$ as

$$(1-\phi)\Delta P = -\zeta^{\mathrm{ve}}\mathcal{C}' + (1-\phi)\Delta P_{dl}, \tag{B4}$$

where $(1-\phi)\Delta P_{dl} = \zeta^{\mathrm{ve}}\mathcal{C}^{\mathrm{K}}$ can be considered a dilatancy pressure. If dilatancy occurs during plastic failure ($\mathcal{C}^{\mathrm{K}} \neq 0$), then $\Delta P_{dl} \neq 0$. Similar to $\zeta_{\mathrm{eff}}$, $\Delta P_{dl}$ is calculated after constraining stresses on the yield criteria and is held constant when solving force-balance equations for strain rates. This constant is calculated by

$$(1-\phi)\Delta P_{dl} = (1-\phi)\Delta P + \zeta^{\mathrm{ve}}\mathcal{C}', \tag{B5}$$





which is always a finite value. Thus, the new formulation using the parameter $\Delta P_{dl}$ resolves the degeneration issue in the effective viscosity approach.

## Appendix C: Full system of equations for the poro-VEVP model

We list the full system of equations for the poro-VEVP model. Details on its development and implementation can be found in Li et al. (2023). Note that the new formulation of $\Delta P$ and the tensor-form permeability are employed in the equations below.

The system of conservation and porosity-evolution equations is

$$-\boldsymbol{\nabla} p^l + \boldsymbol{\nabla} \cdot (2\eta_{\text{eff}}\dot{\boldsymbol{\varepsilon}}') + \boldsymbol{\nabla}\left(\zeta^{\text{ve}}\mathcal{C}'\right) - \boldsymbol{\nabla}\left[(1-\phi)\Delta P_{dl}\right] - \phi\Delta\rho\boldsymbol{g} = \mathbf{0}, \tag{C1}$$

$$\boldsymbol{\nabla}\cdot\boldsymbol{v}^s - \boldsymbol{\nabla}\cdot\left[\boldsymbol{M}_\phi\cdot\left(\boldsymbol{\nabla} p^l + \Delta\rho\boldsymbol{g}\right)\right] = 0, \tag{C2}$$

$$\frac{\partial(1-\phi)}{\partial t} + \boldsymbol{\nabla}\cdot\left[(1-\phi)\boldsymbol{v}^s\right] = 0, \tag{C3}$$

where the modified deviatoric and isotropic strain rates are,

$$\dot{\boldsymbol{\varepsilon}}' \equiv \frac{1}{2}\left[\left(\boldsymbol{\nabla}\boldsymbol{v}^s + (\boldsymbol{\nabla}\boldsymbol{v}^s)^T - \frac{2}{3}\left(\boldsymbol{\nabla}\cdot\boldsymbol{v}^s\right)\boldsymbol{I}\right) + \frac{(1-\phi)\boldsymbol{\tau}^o}{G_\phi\Delta t}\right], \quad \mathcal{C}' \equiv \boldsymbol{\nabla}\cdot\boldsymbol{v}^s - \frac{(1-\phi)\Delta P^o}{Z_\phi\Delta t}. \tag{C4}$$

Here $\boldsymbol{\tau}^o$ and $\Delta P^o$ are the previous deviatoric stress and overpressure, $\Delta t$ is the time-step size. The dilatancy pressure $\Delta P_{dl}$ is calculated by using Eqs. (B5). The effective viscosity $\eta_{\text{eff}}$ is calculated as $\eta_{\text{eff}} = (1-\phi)\tau_{II}/2\dot{\varepsilon}_{II}$, that $\tau_{II} = \sqrt{\boldsymbol{\tau}:\boldsymbol{\tau}/2}$ and $\dot{\varepsilon}_{II} = \sqrt{\dot{\boldsymbol{\varepsilon}}:\dot{\boldsymbol{\varepsilon}}/2}$. The deviatoric stress and overpressure are constrained by the rate-dependent yield surface that

$$\mathcal{F}(\dot{\lambda}, P_e, \tau_{II}) = \sqrt{\tau_{II}^2 + (C\cos\theta - \sigma_t\sin\theta)^2} - (C\cos\theta + P_e\sin\theta) - \eta^{\text{K}}\dot{\lambda} \leq 0, \tag{C5}$$

where $P_e$ is the effective pressure transiting from Terzaghi's stress ($\Delta P = P^l - P^s$) to the full solid stress ($P^s$) at small porosity,

$$P_e = \Delta P + [1 - \exp(-\phi_c/\phi)]P^l, \quad c_{dl} = \exp(-\phi_c/\phi). \tag{C6}$$

Here, $P^l$ is the full liquid pressure taking into account of static pressure. We choose $\phi_c = 10^{-6}$.

The plastic modifier $\dot{\lambda}$ is defined associated with plastic potential $Q$ that

$$\dot{\boldsymbol{\varepsilon}}^{\text{K}} = \dot{\lambda}\frac{\partial Q}{\partial\boldsymbol{\tau}}, \quad \mathcal{C}^{\text{K}} = -\dot{\lambda}\frac{\partial Q}{\partial P_e}. \tag{C7}$$

Here $Q$ is defined as

$$Q(P_e, \tau_{II}) = \sqrt{\tau_{II}^2 + (C\cos\theta - \sigma_t\sin\theta)^2} - (C\cos\theta + c_{dl}P_e\sin\theta). \tag{C8}$$



The dilatancy coefficient $c_{dl}$ is defined in Eq. (C6).

For generality, we still keep the Maxwell shear and bulk viscous terms and its porosity-dependent relation in the computational framework as

$$\eta_\phi = \eta_0 \exp(-27\phi), \quad \zeta_\phi = \eta_0/\phi. \tag{C9}$$

Here we choose $\eta_0 = 10^{30}$ Pa s. For numerical stability, we limit their variation range as $\eta_\phi \geq 10^{-3}\eta_0$ and $\zeta_\phi \leq 10^3\eta_0$. With this choice of parameter, the minimum shear Maxwell time is extremely large, $\eta_0/G \sim 10^9$ years, compared to the simulation time ($< 10^4$ years). The compaction Maxwell time has a similar magnitude too. Therefore, it is essentially a poro-elastic-

viscoplastic rheology in this way.

**Appendix D: Energy analysis of the poro-VEVP model**

This sections explains the calculation of mechanical work rates in the poro-VEVP model associated with different rheological component of the solid phase. Then it discusses the condition that the viscous work in the Kelvin viscoplastic component is negligible.

**D1    Local work rates**

The local work rate associated with deformation at a point can be expressed as the product of the strain rates and effective stresses causing the deformation (Batchelor, 2000; Katz, 2022). In the poro-VEVP model, the local work rate is given by

$$\dot{w} = \bar{\boldsymbol{\sigma}}^{\text{eff}} : \dot{\boldsymbol{e}}, \tag{D1}$$

where the effective stress and strain rates can be decomposed into isotropic and deviatoric parts,

$$\bar{\boldsymbol{\sigma}}^{\text{eff}} = -(1-\phi)\Delta P \boldsymbol{I} + (1-\phi)\boldsymbol{\tau}^s, \quad \dot{\boldsymbol{e}} = \mathcal{C}\boldsymbol{I} + \dot{\boldsymbol{\varepsilon}}. \tag{D2}$$

Here, $\mathcal{C}$ and $\dot{\boldsymbol{\varepsilon}}$ denote the isotropic (compaction rate) and deviatoric strain rates, respectively.

Substituting Eq. (D2) into Eq. (D1) and regrouping with respect to deviatoric and isotropic deformation (cf. Katz (2022)), we obtain

$$\dot{w} = (1-\phi)\boldsymbol{\tau}^s : \dot{\boldsymbol{\varepsilon}} - (1-\phi)\Delta P \mathcal{C}. \tag{D3}$$

The strain rates can be further decomposed into viscous, elastic, and viscoplastic components

$$\dot{\boldsymbol{e}} = \dot{\boldsymbol{e}}^{\text{v}} + \dot{\boldsymbol{e}}^{\text{e}} + \dot{\boldsymbol{e}}^{\text{K}} = (\mathcal{C}^{\text{v}} + \mathcal{C}^{\text{e}} + \mathcal{C}^{\text{K}})\boldsymbol{I} + (\dot{\boldsymbol{\varepsilon}}^{\text{v}} + \dot{\boldsymbol{\varepsilon}}^{\text{e}} + \dot{\boldsymbol{\varepsilon}}^{\text{K}}) \tag{D4}$$





Consequently, the local work rate can also be decomposed into viscous, elastic, and viscoplastic components,

$$\dot{w} = \dot{w}^{\mathrm{v}} + \dot{w}^{\mathrm{e}} + \dot{w}^{\mathrm{K}}. \tag{D5}$$

Each term on the right-hand side includes contributions from both deviatoric and isotropic terms. For example,

$$\dot{w}^{\mathrm{K}} = \bar{\boldsymbol{\sigma}}^{\mathrm{eff}} : \dot{\boldsymbol{e}}^{\mathrm{K}} = (1-\phi)\boldsymbol{\tau}^s : \dot{\boldsymbol{\varepsilon}}^{\mathrm{K}} - (1-\phi)\Delta P \mathcal{C}^{\mathrm{K}}. \tag{D6}$$

### D2 Viscoplastic viscous dissipation energy

In the poro-VEVP model, a Kelvin viscous element with viscosity $\eta^{\mathrm{K}}$ is introduced to regularise the computation of plastic deformation. It increases the total stress of the viscoplastic body by a rate-dependent overstress while sharing the same strain rates as the plastic element. Therefore, the dissipation rate of the viscoplastic component can be decomposed as

$$\dot{w}^{\mathrm{K}} = \dot{w}^{\mathrm{p}} + \eta^{\mathrm{K}}\dot{\boldsymbol{e}}^{\mathrm{K}} : \dot{\boldsymbol{e}}^{\mathrm{K}}, \tag{D7}$$

where the first and second terms on the right-hand side represent the Kelvin plastic and Kelvin viscous dissipation energy, respectively.

Comparing this equation with Eq. (D6), we find that the Kelvin viscous term is negligible if $\|\eta^{\mathrm{K}}\dot{\boldsymbol{e}}^{\mathrm{K}}\| \ll \|\bar{\boldsymbol{\sigma}}^{\mathrm{eff}}\|$. In the tensile failure regime, the magnitude of effective stress is about the similar size to the tensile strength when the Kelvin viscosity is sufficiently small, that is $\|\bar{\boldsymbol{\sigma}}^{\mathrm{eff}}\| \sim \sigma_t$. Therefore, the condition for negligible Kelvin viscosity can be written as

$$\eta^{\mathrm{K}} \ll \frac{\sigma_t}{\|\dot{\boldsymbol{e}}^{\mathrm{K}}\|}. \tag{D8}$$

We use preliminary computations to extract $\|\dot{\boldsymbol{e}}^{\mathrm{K}}\|$ and then estimate the conditions for $\eta^{\mathrm{K}}$. The maximal plastic strain rate is higher when the propagation rate is faster. In a computation that has $v_t \sim 7$ m/yr, $\|\dot{\boldsymbol{e}}^{\mathrm{K}}\| < 10^{-10}$ s$^{-1}$. Taking $\sigma_t = 1.25$ MPa, we find $\eta^{\mathrm{K}} \ll 10^{16}$ Pa s. A sensitivity test to the value of $\eta^{\mathrm{K}}$ can also confirm whether the effect of Kelvin viscosity is negligible.

In this manuscript, we choose $\eta^{\mathrm{K}} = 10^{10}$ Pa s which is sufficiently small for all cases considered.

### Appendix E: Stress distribution of the (poro-)LEFM model

The stress distribution at the tip of the poro-LEFM model is identical to the mode-I fracture of the LEFM model,



$$
\begin{Bmatrix} \sigma_{xx} \\ \sigma_{zz} \\ \sigma_{xz} \end{Bmatrix} = \frac{K_c}{\sqrt{2\pi r}} \begin{Bmatrix} \cos(\vartheta/2)[1 - \sin(\vartheta/2)\sin(3\vartheta/2)] \\ \cos(\vartheta/2)[1 + \sin(\vartheta/2)\sin(3\vartheta/2)] \\ \sin(\vartheta/2)\cos(\vartheta/2)\cos(3\vartheta/2) \end{Bmatrix},
\tag{E1}
$$

where $r$ and $\vartheta$ are the polar coordinate system from the fracture tip, as shown in Fig 5 (a). This formulation has also been used in Li et al. (2023).

**Appendix F: Stresses and pressure inside of the dyke**

This section discusses the differences in stresses and pressure inside the dyke between the poro-VEVP and poro-LEFM models. Taking $p^l$ (liquid pressure) at the tail as an example, we have $p^l = 0$ and $\partial p^l/\partial z = 0$ in the poro-LEFM model, but non-zero

values for both in the poro-VEVP model. These differences stem from the nature of geometry and the complexity of stress balances.

Firstly, the poro-LEFM model assumes an infinitely long dyke, while the poro-VEVP model cannot make such an assumption. Consequently, the far-field condition of zero pressure and pressure gradient can be applied directly to the poro-LEFM model, but not to the poro-VEVP model.

Secondly, the stress balance in the poro-LEFM model is simpler than in the poro-VEVP model. The poro-LEFM model assumes $p^l = p^s$ and takes $p^s$ as an elastic stress of the solid phase under one-dimensional deformation, as shown in Eq. (A7). However, the poro-VEVP model has a two-dimensional force-balance equation involving the gradient of tensor-form solid stresses and an extra term of static pressure gradient $\phi\Delta\rho\boldsymbol{g}$, as shown in Eq. (6).

This complexity is evident in the force balance equation along the dyke, which is the z-component of Eq. (6),

$$
-\frac{\partial p^l}{\partial z} + \frac{\partial}{\partial x}[(1-\phi)\tau_{xz}^s] + \frac{\partial}{\partial z}[(1-\phi)\tau_{zz}^s] - \frac{\partial}{\partial z}[(1-\phi)\Delta P] + \phi\Delta\rho g = 0.
\tag{F1}
$$

Here, $\tau_{xz}^s$ and $\tau_{zz}^s$ are components of the tensor-form solid deviatoric stresses, and $\Delta P$ is the compaction stress. These stresses are associated with deformation in both $x$ and $z$ directions. Even assuming no solid deformation, we have $\partial p^l/\partial z = \phi\Delta\rho g$, where liquid pressure balances with static pressure. In general, none of the terms in the equation can be eliminated through scaling analysis.

Figure F1 shows numerical results of the vertical distribution for all five terms in the equation above for the reference case at $t = 2$ kyr. Sufficiently far from the tip, all terms become invariant with respect to their vertical position, and none can be considered zero. Therefore, $p^l$ is coupled with the gradient of full tensor-form stresses of the solid phase, and thus also the full tensor-form strain rates. These values can only be determined through numerical computation, preventing us from prescribing boundary conditions consistent with the supposed stress gradient in the tail. This unavoidable difference leads to a boundary

layer at the bottom serving as a transition in the numerical results, as shown in Fig. 3(b–c).





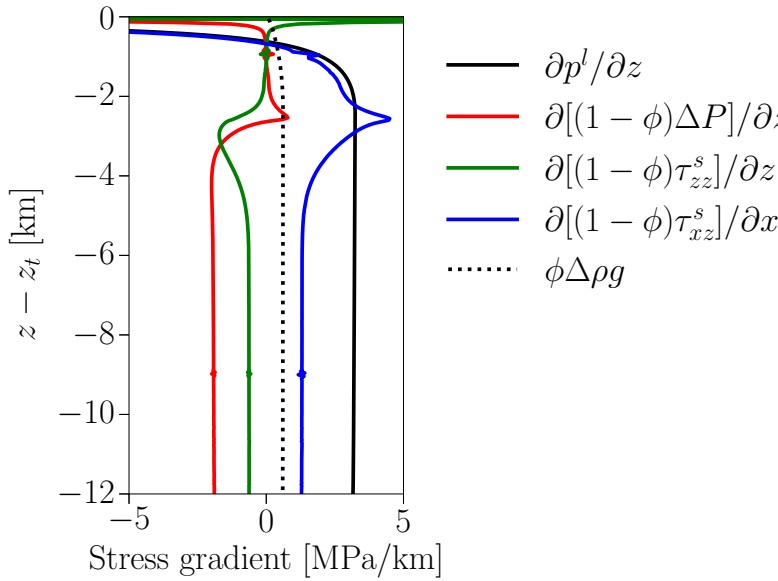

**Figure F1.** Components of vertical stress gradients of the reference case at $t = 2$ kyr.

Hence, quantitatively comparing stresses inside the dyke between the poro-VEVP and poro-LEFM models may not be reasonable, as evidenced by the mismatch in the grey region in Fig. 5(b). However, we can compare stresses in the zero-porosity region outside the dyke (Fig. 5), where both models describe a two-dimensional elastic stress distribution associated with the tip fracture. The poro-LEFM model's 2D stress field components are shown in Eq. (E1), representing a toughness-dominated distribution. The poro-VEVP model's components are computed numerically, with the dominant rheology being elasticity and the strong plastic deformation at the tip qualitatively similar to a discrete fracture. This intense plastic deformation is seen as the abrupt peak of plastic dissipation energy in Fig. 3 (d).

We also observe similarity in the porosity distribution inside the dyke near the tip (Fig. 4(e)), implying similar $\partial p^l/\partial z$ near the tip due to Darcy's equation. Figure F1 shows $\partial p^l/\partial z$ can be a leading term in the force-balance equation near the tip, suggesting similar fracture-dominated deformation despite different far-field stresses in the poro-VEVP dyke.

*Author contributions.* All authors actively contributed to the project through regular meetings and provided critical feedback. YL developed and implemented the method, made the analysis and visualizations. YL and TD developed the codes for the poro-LEFM model. YL and AP developed the codes for the poro-VEVP model. RK acquired the funding and supervised the project. YL and RK wrote the paper. TD and AP provided critical feedback on the writing. All authors revised the final version of the paper.

*Competing interests.* The authors have no competing interest to declare.



*Acknowledgements.* This research received funding from the European Research Council under Horizon 2020 research and innovation program grant agreement 772255. AP acknowledges support from the Royal Society (URF\R1\231613). Numerical simulations were computed on the Arcus-C cluster from the Advanced Research Computing (ARC) services at the University of Oxford.



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
