# Peer review of "Models of buoyancy-driven dykes using continuum plasticity or fracture mechanics: a comparison"

_EGUsphere, 2024_

## Author Response (AR1)

**Manuscript "Models of buoyancy-driven dykes using continuum plasticity and fracture mechanics: a comparison" – Response to CC1**

Comments to the Author(s):

Very good multidisciplinary research with a focus on dykes, I definitely enjoyed reading it. The amount of literature on the hydraulic properties of fractured rocks at such a large scale is not large. Please, see my specific comments to improve the manuscript.

**R:** We thank the reviewer for this feedback. Below, we address each specific comment in turn, with our responses indicated by **R:**.

- Lines 62-70. Porosity. Total or effective porosity? When I think to the porosity of fractured rocks, I tend to consider this concept. Think if you need to specify something.

  **R:** In this manuscript, we assume total and effective porosity are equivalent, implying that all voids within the grain matrix are interconnected and fully saturated with liquid. We acknowledge that a distinction between these porosities exists and is important in some contexts. However, for partially molten rock in the asthenosphere, the pore structure is controlled by textural equilibrium, which results in a highly connected pore network. Therefore, equating the total and effective porosity is common practice in the geodynamic modelling community for the scales and processes considered. Thus, further elaboration is not included here.

- Lines 10, 71-76. Permeability anisotropy. Vertical or horizontal? You need to explain this point in the abstract.

  **R:** To clarify, we will revise the sentence in line 73 as follows:

  Original: "We resolve this discrepancy by introducing an anisotropic permeability tensor into the poro-VEVP model..."

  Revised: "We resolve this discrepancy by introducing an anisotropic permeability tensor into the two-dimensional poro-VEVP model..."

  This revision makes clear that the anisotropic permeability is implemented within a 2D model, which is later described as occupying a vertical plane.

- Line 73. Permeability tensor. Are you thinking to apply this concept at which observation scale?

  **R:** The "observation scale" is not a defined quantity in the context of our mathematical model. We define anisotropic permeability at the subgrid scale, where it is described by the tensor. Hence the anisotropy is uniform within a grid cell; anisotropy at larger scales emerges as part of the numerical solution. This large scale structure might be compared with observations.

- Line 73. Permeability tensor. If you think that is a good idea to discuss the observation scale, please provide details on the depth and lateral extension.

  **R:** Thank you for the suggestion. However, we opt not to discuss this topic, as it is less relevant to the manuscript's main focus.

- Lines 74-75 "Anisotropic permeability can arise from anisotropic stresses and aligned pores or fractures". Please, insert recent and relevant literature on the anisotropic permeabilities due to either anisotropic stress and orientation of fractures:

  - Medici G, Ling F, Shang J 2023. Review of discrete fracture network characterization for geothermal energy extraction. Frontiers in Earth Science, 11, 1328397.

  - Lei, Q., Latham, J.P. and Tsang, C.F., 2017. The use of discrete fracture networks for modelling coupled geomechanical and hydrological behaviour of fractured rocks. Computers and Geotechnics, 85, 151-176.

  **R:** Thank you for suggesting these relevant studies. We will include them in the references of the revised manuscript.

- Lines 174. You reference multiple times Snow 1979. Do you need to discuss the hydraulic / mechanical aperture of the joints. What about the cubic law?

  **R:** We do not need to discuss the hydraulic aperture of the joints and its effect on permeability models as Snow (1979) did. This is a subgrid feature that we cannot resolve. Such a detailed discussion lies beyond the scope of this manuscript, which focuses on comparing the continuum (poro-VEVP) and discrete (LEFM) models given the same constitutive laws.

- Figure 1. Do you need to insert a spatial scale in your conceptual model?

  **R:** We appreciate the suggestion but believe a spatial scale is unnecessary for this figure. The schematic is designed to highlight differences between the classical LEFM model and the proposed poro-LEFM model. For such a conceptual diagram, it is not necessary to either show dimensional sizes or the relative size ratio, such as the head size to the far-field aperture. A similar sketch is commonly used in LEFM studies, for example Figure 1 in Roper and Lister (2007).

  - Roper, S. M. and Lister, J. R., 2007. Buoyancy-Driven Crack Propagation: The Limit of Large Fracture Toughness, Journal of Fluid Mechanics, 580, 359–380.

- Figure 3a. The figure describes the porosity and solid deformation field at t = 2 kyr. This is an important figure and if the reader wants to catch the details need to zoom in a lot. Please, enlarge the size.

  Figure 3a. Can this figure be separated from the others?

**R:** Thank you for this suggestion to improve the presentation of results. An animation is included in the supplement to provide a clearer visualisation of the evolution of both porosity and the solid deformation field.

- Figure F1. This is a very important figure from a conceptual point of view. Indeed, the image shows physical variations as a function of the depth. If you introduce this figure in the main body of the manuscript, you would rise either the readability or the impact of your research.

  **R:** We appreciate this suggestion. However, after careful consideration, we believe it is more appropriate to leave Figure F1 in the Appendix. This figure highlights a complex difference that, while interesting, has limited impact on the main conclusions and is not yet fully understood. As such, we include it as supplementary material for specialist readers without diverting focus from the primary objectives of the manuscript.

**Manuscript "Models of buoyancy-driven dykes using continuum plasticity and fracture mechanics: a comparison" – Response to RC1**

We are grateful for this thorough review. As the reviewer correctly noted, there were three typos: the omission of the factor of 1/3 in front of the isotropic strain rate components in Equations D2 and D4, and an incorrect expression, $\Delta P = P^l - P^s$, on the line below Equation C5, which should have been $\Delta P = P^s - P^l$. We have corrected these errors in the revision (the code implementation was correct in the first instance).

Many of the other comments appear to stem from a misunderstanding of the scope of this manuscript. This manuscript has one specific and limited aim: to compare the poro-VEVP model, published in Li et al. (2023), against the classical LEFM model for a buoyancy-driven dyke. To best focus on this key purpose, we made the following choices:

(i) We do not review in detail other models that include plasticity into a poro-(visco)elastic framework.

(ii) For consistency with Li et al. (2023), we use the same terminology and notations.

(iii) To avoid repetition, we omit mathematical details explained in Li et al. (2023), and recapitulate only the key elements of the mathematical model in the Appendices.

Regarding (i): reviewing other plastic models (except for those used to simulate dyking) is beyond the scope of this manuscript. Therefore, we initially kept the literature review brief, covering only the most relevant studies. In response to the review and to better connect our ideas with related ones, we have incorporated some of the suggested references where appropriate. However, we do not discuss them in detail to avoid distraction.

Regarding (ii) and (iii), the reviewer has made valuable suggestions and highlighted places where different approaches have led to different notations. However, it is not our aim to address these broad issues in the literature. We feel that the current formalism and notation best serves our admittedly narrow aims and maintains consistency with our closely related paper. Readers seeking to understand the workings of the poro-VEVP model are encouraged to read Li et al. (2023).

The remainder of this response is organised into two sections based on the type of response. In the next section, we discuss revisions to improve the manuscript; in the section after that we provide further clarifications for the reviewer.

**1    Revise manuscript for improvement**

- Line 12. "Magmatic dykes, formed by fluid-driven fracture, are important pathways for magma ascent across the lithosphere." This is just one of the hypotheses; correct the sentence and cite other possible mechanisms.

**R:** This sentence is not incorrect. Certainly we agree that there are other possible mechanisms. However, reviewing all possible mechanisms does not serve the aims of this manuscript. We rephrase the sentence as "Magmatic dykes, formed by fluid-driven fracture, are an important pathway for magma ascent across the lithosphere." This more clearly indicates that they are one mechanism out of a larger set.

- Line 105. This equation is not standard, give explanations.

  **R:** This equation is a slight modification of the standard formulation in the LEFM model. We added further explanation in Line 111: "Note that Eq. (2) adapts the standard LEFM elastic stress formulation (e.g., Lister (1990)) to account for porosity effects on solid deformation."

- Lines 169-170. "This approach only changes how the stress-balance equation is linked with the plastic yield condition, without altering either of the two physics." Explanation is confusing at best. I do not see how physics stays the same.

  **R:** We removed this sentence to avoid confusion. The phrase 'without altering either of the two physics' was intended to clarify that neither the stress-balance equation nor the plastic yield condition was changed. Evidently, it was not clarifying.

- Line 209. Since you do not provide your rheology equations where G and Z are parameters, it is useless to provide these expressions here.

  **R:** We moved the expressions for $G_\phi$ and $Z_\phi$ to Table 1, and rephrased the paragraph in Lines 208–210 as "The elastic shear ($G_\phi$) and bulk ($Z_\phi$) moduli follow porosity-dependent relations, as shown in Table 1. Note that ...".

- Line 220. It would be helpful with the 2D figure illustrating the initial distribution of porosity.

  **R:** We will upload an animation of the porosity field evolution to illustrate dyke propagation. This animation will demonstrate the initial porosity distribution.

- Line 292. "the tip advances approximately the same distance in each 0.4 kyr interval" Why? Discuss mechanisms and reasons for such behavior.

  **R:** This is certainly an interesting question. A constant propagation speed is a result of stresses reaching an equilibrium and steady state in a reference frame moving at the speed of the tip. Constant speed is essential for a valid comparison with the LEFM model. However, we do not fully understand the mechanisms responsible for this equilibrium. Our approach was to configure the boundary conditions of the poro-VEVP model to mimic a buoyancy-driven dyke in the LEFM model. In the LEFM model, stress equilibrium is assumed, resulting in constant-speed dyke propagation. In the poro-VEVP model, equilibrium is achieved as a result of the boundary conditions and domain configuration.

To clarify this in the manuscript, following this sentence, we added: 'It implies that a stress equilibrium has been achieved, which is consistent with the assumptions of the LEFM model'.

- Lines 372-373. "The dyke width is determined by the grid size, which is a limitation of the present discretized solutions of the continuum models." Discuss these limitations and ways to overcome them. The problem of mesh dependency and respective regularization is widely discussed in the geodynamic community (e.g., Duretz et al., 2023).

  **R:** We have added a section to the Appendix discussing mesh dependency and refer to this new section for details. We carefully considered the Duretz regularisation in our previous work. While that regularisation works well for shear failure, it appears to be inadequate for tensile failure. This is an important issue, but one that we do not currently know how to resolve.

- Lines 539-541. "For generality..." Equations (C9) are not general. Where do they come from?

  **R:** We change "For generality, we still keep the Maxwell shear and bulk viscous terms and its porosity-dependent relation in the computational framework" to read "Although the models in this paper are dominantly elastic and plastic, we retain viscosity in our formulation for generality. We employ the following porosity-dependent relationships for the Maxwell shear and bulk viscosity,".

**2 Further clarification to the reviewer**

- "First, they had to introduce a strong anisotropy of permeability inside a channel they produced, and the flow was essentially driven by buoyancy and this anisotropy. Thus, dyke always follows the direction of the anisotropy. However, most of the rocks have higher permeability in the horizontal direction, and the planes of weakness of anisotropic rocks are also in the horizontal direction. Thus, this argument would be more suitable to simulate sills. In reality, sills and dykes often accompany each other forming interconnected plumbing systems. "

  **R:** We wish to clarify two points of misunderstanding. First, we did introduce a strong anisotropy of permeability; however, this was designed to reduce lateral fluid leakage from the dyke, leading to a better match with the LEFM model. It does not drive the flow, and its direction is a result of the PDEs, not a prescribed parameter (noting, however, that in this manuscript we only allow the direction being either horizontal or vertical). Second, we did not model the background anisotropic permeability of sedimentary rocks, although this is certainly an interesting topic. This

manuscript focuses on comparing two theoretical models and therefore employs the simplest possible geological features.

- "Second, they "prescribe" dyke by setting the initial elevated porosity region at the bottom of their computational domain prescribed by anisotropic Gaussian porosity distribution."

  **R:** We did not prescribe a dyke; rather, dykes emerge as a feature of the numerical solutions of the PDEs. Any initial porosity distribution could lead to one or more similar dykes. However, we chose this particular initial distribution to promote a convenient comparison with the LEFM dyke, which is the focus of this manuscript.

- "And third, the width of their dyke is one grid size. This would suggest that at higher resolution, their dyke will be an infinitely thin line. In reality, dykes always have a finite length, sometimes meters. "

  **R:** We are aware of the limitation that the dyke width is determined by the grid size. This limitation was previously discussed only in the Discussion section. In the revision, we also further discuss grid dependency in a new appendix (Appendix E).

- Lines 71-74. "Isotropic permeability within the poro-VEVP dyke promotes widening by horizontal porous flow, a behaviour not associated with real (or LEFM) dykes. We resolve this discrepancy by introducing an anisotropic permeability tensor into the poro-VEVP model to limit leakage and enhance fracture propagation..." Discuss other possible reasons for not reproducing dyke. For example, an underpressure at the fracture tip generated at the moment of fracture opening can be a driving force for the flow. Fracture opening must be must faster process than Darcy flow, and thus, there would be a region with undrained conditions at the tip where pore volume is increased due to fracture generation that would lead to the pressure drop at the tip, increasing pressure gradient and thus the flow.

  **R:** Other factors may contribute to the widening of the poro-VEVP dyke over time. However, identifying one solution to this issue is sufficient for the purpose of this manuscript. This comment suggests an interesting alternative, which, however, requires examination in future work.

- Line 108. Why do you assume fluid pressure is equal to solid pressure? Further in the text you assume that pl = const and arrive at equation (5), while ps given by equations (2) and (3) is not constant. These two assumptions are not consistent.

  **R:** Here $p^l = p^s$ assumes stress balance between the liquid and solid. This is a common assumption in the LEFM model and does not lead to the violation suggested by the reviewer. Equation (2) is the equation for the solid elastic stress. Equation (3) is the fracture criterion at the tip. A constant $p^l$ or $p^s$ at the far field is a boundary condition. Eq (2) is consistent with this boundary condition when $\partial \phi h / \partial z = 0$ at

the far field, which implies a constant-width or constant-porosity assumption at the far field. A dyke with a constant width (porosity) at the far field is the case that we use to compare the LEFM and poro-VEVP model in this manuscript.

- Line 147. Equation (6) and the corresponding equation in the Appendix have a wrong gravity term. The conservation of total momentum has a total density in the gravity term (Bercovici et al., 2001; Yarushina and Podladchikov, 2015).

  **R:** The gravity term here, and the corresponding equation in Appendix C (equation C1), is correct. This term, $-\phi\Delta\rho\boldsymbol{g}$, is derived by subtracting a gradient of static pressure associated with the solid phase, $\rho^s\boldsymbol{g}$, from the gravity term with a total density, $\bar{\rho}\boldsymbol{g}$. Correspondingly, the pressure term becomes $p^l$ representing the dynamic liquid pressure, consistent with the definition in section 2.1. This formulation is used in many references cited by this manuscript, and is consistent with the formulation suggested by the reviewer (i.e., Bercovici et al., 2001; Yarushina and Podladchikov, 2015).

- Line 248. "Appendix D provides details of the formulation for each local work rate." Incorrect. You provide this estimation only for one in the appendix.

  **R:** We explain the details of each mechanism up to the last step. For the last step, we use the plastic work rate as an example to show the final expression. We expect that readers can work out the similar expressions for viscous and elastic components without difficulty.

- Line 260. Why propagation speed is assumed constant?

  **R:** This is the case we simulate in both the poro-VEVP and LEFM models to compare them. While Equation (19) could certainly be generalized to accommodate a variable speed, such an extension is beyond the scope of this manuscript.

- Line 296-299. The 2D figure of plastic failure around the tip would help in understanding. You are referring everywhere in the text that your dyke is generated by mode I fracture. How can you be sure that it is mode I and not mode II mode, given that your yield criterion contains both? This statement needs further proof and illustration. Are there any shear stresses associated with dyke propagation?

  **R:** The LEFM solution to which we compare is a mode-I solution. In Li et al. (2023), we demonstrated quantitative agreement between this type of tensile failure and mode-I fracture in LEFM theory by comparing the stress field around the tip. In this manuscript, we further support this agreement by recognising the equivalence between the mode-I fracture energy in the LEFM model and the plastic-dissipation energy in the poro-VEVP model. However, we agree that completely isolating mode-I and mode-II fractures is impossible because the yield criterion incorporates both shear and compaction stresses. Distinguishing between shear and tensile failure is

not truly feasible when using a smooth yield surface. Energetically, both modes contribute to the total plastic dissipation. Currently, we continue to use the definition from our previous work, where plastic failure associated with negative $\Delta P$ is categorised as tensile failure.

- Lines 418-422. Maybe anisotropy is not the right mechanism? The results do not look convincing.

  **R:** We apologise for being unable to address this comment fully due to its lack of detail. It does not specify which results are unconvincing or the reasons why. Lines 418-422 only discuss the limitations of the specific form of anisotropic permeability chosen for this manuscript.

- Line 526. What is overpressure and dilatancy pressure here? Consider using conventional terminology. You already used equation B5 when deriving C1 from (6). You need a separate independent equation for it.

  **R:** Overpressure is a commonly used name for $\Delta P$. Dilatancy pressure $\Delta P_{dl}$ is a new term introduced in a new formulation for $\Delta P$, presented in Eq. (8). It represents a correction term for compaction stress due to dilatancy after plastic failure. We believe this terminology is clear and should not cause confusion.

  Our system of equation is mathematically consistent and presents no issue as suggested by this comment. This approach is similar to the conventional effective viscosity approach, but replacing the parameter $\zeta_{\text{eff}}$ with $\Delta P_{dl}$. For details on the effective viscosity approach, we suggest to read Appendix D in Li et al. (2023) and references therein.

- Lines 551-552. "The local work rate associated with deformation at a point can be expressed as the product of the strain rates and effective stresses causing the deformation (Batchelor, 2000; Katz, 2022)." The concept of work rate is much more fundamental and dates much earlier than 2000 or 2022. Consider more appropriate references here.

  **R:** These references are not intended to acknowledge the original source; indeed, this concept is fundamental to continuum mechanics. We cite two books here: Batchelor's as a classic textbook on fundamental fluid dynamics, and Katz's, to connect this concept to the two-phase geodynamic contexts.

- Line 570-576. Equations (D7) and (D8) need further explanations. You assumed that for viscoplastic stresses. However, this is not what is written as a viscoplastic constitutive equation (C7). Provide the proper derivation.

  **R:** We believe that Equations (D7) and (D8) are sufficiently explained in the context. This comment suggests that the reviewer interprets $\eta^{\text{K}}\dot{e}$ as representing the

total viscoplastic stresses; however, this term represents only the stress for the Kelvin(viscoplastic) viscous component. We clarified the meaning of the two terms in Equation (D7). Readers are expected to understand that each Kelvin component in a viscoplastic material experiences different stresses but identical strain rates.

Manuscript "Models of buoyancy-driven dykes using continuum plasticity and fracture mechanics: a comparison" – Response to RC2

**1    General comments**

In general, I believe that this work is really interesting and should be definitely published after a moderate revision. Below I summarize a few discussion topics that should be addressed in the manuscript.

**R:** We thank the reviewer for this positive feedback. Below, we address each specific comment in turn, with our responses indicated by **R:**.

Before addressing the comments, we clarify that this manuscript has one specific aim: to benchmark the poro-VEVP model, published in Li et al. (2023), against the classical LEFM model. This poro-VEVP model (arguably) showed an approach to simulate dyking in a continuum model. However, the analysis by Li et al (2023) did not convincingly resolve all issues for this purpose. The present manuscript shows that this poro-VEVP model does achieve a substantial similarity to the dyke in the classical LEFM model.

**2    Specific comments**

- The results of poro-VEVP models presented in Section 3.1 demonstrate that dyke is essentially locked to one grid cell across the width. The text only discusses the mesh dependence issues in the context of boundary conditions. However I believe that it's not the only factor that is important in this context. What do you think the intrinsic length scale for the dyke width in poro-VEVP formulation should depend on? What happens if a theoretical dyke width is infeasible to resolve in the numerical model, e.g. in 3D? How would you parametrize such a case? A series of models with varying lateral resolution would provide a valuable insight in this issue.

  **R:** We have added Appendix E, including a new plot (Figure E1), to illustrate the effect of lateral resolution. This figure shows different results for mesh-dependency tests with either a fixed flux or a fixed flow rate. This difference arises because the dyke width cannot be smaller than one grid cell; thus, the same flow rate leads to different fluxes when varying the lateral resolution. Even with a fixed flux, the plastic dissipation rate is also affected by the lateral resolution.

  However, the propagation speed can be made independent of the cell size by prescribing a fixed flux. This regularisation might originate from Darcy's law, but it is unclear whether this introduces an intrinsic length or time scale. Further studies are certainly required.

Regarding simulating a dyke in 3D, we are unsure how to parameterise such a case to reduce computational cost.

- Slow propagation speed in poroVEVP models was mainly attributed to a relatively large Darcy drag in the fracture. I think this conclusion is generally correct. It was also correctly pointed out that propagation speed can be increased by larger permeability enchantment parameter. The question is what values should be actually used? Perhaps the power-law porosity-permeability dependence is just wrong for the dyke case, or the power-law exponent should be larger. Here is my constructive suggestion: we can probably somehow estimate the effective permeability in the dyke such that Darcy drag would be made equal to the viscous drag of Poiseuille flow under the same fluid pressure gradient.

  **R:** We have shown the effective permeability at which the Darcy drag equals the viscous drag of Poiseille flow, albeit indirectly, in Figure 2a. This figure demonstrates that the speed of a poro-LEFM model approaches the speed of an LEFM model as $\phi \to 1$, when the relationship between dyke width, $h_0$, and mobility prefactor, $M_0$, satisfies $h_0 = (12 M_0 \mu)^{1/2}$. This relationship is derived using the cubic power-law porosity-permeability relation. Assuming a constant-width poro-VEVP dyke, this relationship can be used to estimate the effective permeability in that model.

- Despite that reliable analysis of mechanical energy dissipation around the dyke tip was performed, it is still not guaranteed that dyke propagation speed in poro-VEVP models is completely independent of the mesh resolution. It would be again insightful to run a series of models with different resolution to investigate how this affects the propagation speed.

  **R:** We added Appendix E and Figure E1 to show that the dyke propagation speed is independent of the mesh resolution.

- The viscous rheology currently used in the poro-VEVP models is only porosity-dependent. Would it take a significant effort to incorporate a stress-dependent viscous rheology e.g. power law? For example a complete redesign of the iterative point-wise stress computation algorithm would be necessary in this case.

  **R:** While this is beyond the scope of this manuscript, we offer a brief comment. The poro-VEVP model incorporates two viscous components: the Maxwell and Kelvin viscous terms. We assume this comment refers to the Maxwell viscous term, as there is currently no physical justification for employing such a complex rheology for the Kelvin viscous term. Incorporating a stress-dependence into the Maxwell viscous component is straightforward and does not require any modification of the point-wise stress computation algorithm. In the momentum equation, this is implemented as a non-Newtonian viscosity dependent on strain rates. During the point-wise stress

computation, both the strain rates and viscosity are treated as constants. Therefore, the same algorithm remains applicable.

- The smooth plasticity model used in this work raises a few concerns. First of all, hyperbolic approximation for the Mohr-Coulomb yield surface in the meridional plane was initially introduced by Abbo and Sloan (1995), two years earlier than in Carol et al. (1997). Please acknowledge that fact. Second, from the definition of the flow potential it follows that this model does not handle the non-associative case with dilatation angle approaching zero.

  ...

  To summarize, it is quite clear that plasticity model should assume uncoupled description of the shear and tensile failure modes which is not the case for the hyperbolic approximation. Even the original ad hoc model of Carol et al. (1997) is not a good candidate since it is not smooth and not convex (see the first plot). Please discuss this rather important issue in the text.

  **R:** We greatly appreciate this detailed examination of the plasticity model used in Li et al. (2023) and this manuscript.

  We have added citations to the original sources for the hyperbolic yield surface concept where it is first mentioned on page 2: "(e.g., Abbo and Sloan, 1995; Carol et al., 1997)".

  We agree that the plastic potential function in the original Carol model can be non-smooth and non-convex for certain parameter choices. We thank the reviewer for pointing this out. However, this also depends on the choice of function for the dilatancy coefficient. Our poro-VEVP model uses a different function than the Carol model, so under these conditions, this issue may not impair the model's capabilities. We will address this concern further below, demonstrating that it is not a problem in our model.

  This comment also questioned why we chose the dilatancy coefficient to depend on porosity only, as

  $$c_{dl} = \exp\{(-\phi_c/\phi)\}.$$

  We are working with a two-phase model that imposes incompressibility of the individual phases. An essential assumption in this context is that dilatancy can only occur where mobile fluid is present. Solid grains in regions without fluid thus cannot undergo dilatancy. A naive approach to incorporate this assumption is to prescribe a porosity-dependent dilatancy coefficient. The simplest approach is to allow full dilatancy ($c_{dl} = 1$) if sufficient fluid presents, and no dilatancy ($c_{dl} = 0$) otherwise. Computationally, a function to facilitate a smooth transition for $c_{dl}$ is desirable. Therefore we choose this exponential function where $\phi_c$ is a parameter controlling the scale of $\phi$ over which the transition of $c_{dl}$ occurs.

This comment notes that the choice of $\phi_c$ affects $c_{dl}$, with examples of $\phi_c = 10^{-1}$, $10^{-2}$, and $10^{-3}$. While this parameter can indeed impact $c_{dl}$ and thus affect the numerical results, we mitigate this impact by choosing a very small value, $\phi_c = 10^{-6}$. With this choice, $c_{dl} = 0.9$ and $0.98$ when $\phi = 10^{-5}$ and $10^{-4}$, respectively. Therefore, the transition of $c_{dl}$ occurs in the region with porosity three to four orders of magnitude smaller than the porosity in the main region inside of the dyke. Thus, $c_{dl} \approx 1$ in almost all regions with sufficiently large porosity. In other regions with very small porosity, the value $c_{dl}$ does not affect the results in this manuscript because plastic failure does not occur due to the high plastic effective compressive stresses in nearly zero-porosity regions. Because $c_{dl}$ is effectively constant in the regions of interest, our plastic potential function remains smooth and convex in the relevant parameter range.

However, we agree that this plasticity model could be improved to better capture both shear and tensile failure modes within a single framework. In this plasticity model, dilatancy could be related to stresses or shear strain rates, potentially relevant to shear failure in shallow regions (where opening pore space is readily accommodated by inflow of water and/or air, or even density change of air). We intend to explore this possibility in future work.

To further clarify the choice of porosity-dependent dilatancy, we have added the following text to Appendix C: "Note that we choose $c_{dl}$ to depend on porosity. This choice contrasts with the stress-dependent formulation used in [?], which studies cracks in an engineering context. In our model, $c_{dl} \approx 1$ everywhere the porosity is not vanishingly small, and $c_{dl} \approx 0$ in non-porous regions. The exponential function is chosen to provide a smooth transition between these two states. "

---

## Author Response (AR2)

Dear Editors,

Thank you for the opportunity to revise our manuscript, "Models of buoyancy-driven dykes using continuum plasticity and fracture mechanics: a comparison". We appreciate the constructive feedback from you and the reviewer. We have now addressed all the remaining points and believe the manuscript is significantly improved in its clarity and transparency.

In summary, we have undertaken the following revisions as agreed:

- **Reviewer Comments**: All new specific comments from the reviewer have been addressed, with clarifications incorporated directly into the manuscript.

- **New Compaction Formulation**: We have improved the explanation of our new compaction formulation in Section 2.2.1 by clarifying the role of the Picard iteration scheme in handling nonlinearities and have enhanced the cross-reference to our previous work for full details.

- **Plasticity Model Limitations**: We have added a new paragraph to the Discussion section acknowledging the limitations of the employed plasticity model, placing our work in the context of this active area of research.

- **Numerical Implementation**: We have added succinct information about our numerical framework to the introduction and Section 2.2.4 to make the manuscript more self-contained, with clear signposting to Li et al. (2023) for details.

We trust that these revisions address your concerns and have strengthened the manuscript. We look forward to your decision.

Sincerely,

Yuan Li & Co-authors

**Manuscript "Models of buoyancy-driven dykes using continuum plasticity and fracture mechanics: a comparison" – 2nd Revision**

**1 Review's comments**

The revised manuscript shows several improvements and thoughtful clarifications in response to reviewer comments. I appreciate the authors' efforts and recognize their intention to sharpen the focus of the work. However, several substantial concerns from the first review round remain either unaddressed in the manuscript itself or insufficiently justified. Many of the explanations are offered only in the rebuttal, while key modeling assumptions, derivations, and terminology still lack clarity or supporting evidence in the main text. To meet the standards of transparency and reproducibility, these elements must be incorporated directly into the manuscript.

**R:** We thank the reviewer for their constructive feedback. In response to their specific comments, we have revised the manuscript to directly incorporate the necessary explanations of our modeling assumptions, terminology, and methods. We agree that these changes significantly improve the manuscript's clarity and transparency.

**1.1 Specific comments**

- Clarification of scope: The authors themselves note that "many of the other comments appear to stem from a misunderstanding of the scope of this manuscript," and indicate that both reviewers required clarification on this point. This strongly suggests that the scope is not sufficiently defined in the manuscript itself. The clarification provided in the rebuttal should be incorporated into the revised text—preferably early in the introduction—to prevent similar confusion for future readers.

  **R:** We thank the reviewer for this constructive suggestion. We have revised the first paragraph of introduction to clarify that the central goal of this manuscript is to rigorously benchmark our continuum model against linear elastic fracture mechanics (LEFM) as a critical validation step.

- Line 12 – Formation of dykes: A previous comment on Line 12 remains inadequately addressed. The sentence "Magmatic dykes, formed by fluid-driven fracture, are an important pathway for magma ascent across the lithosphere" still implies a singular mechanism, even after the minor rephrasing. To ensure scientific accuracy and neutrality, I recommend splitting the sentence: "Dykes are an important mechanism for magma ascent. They can be formed, among other mechanisms, by fluid-driven fracture." This makes the distinction between the general importance of dykes and the specific process of their formation more precise, without requiring an exhaustive review of alternative mechanisms.

**R:** Changed.

- Anisotropic permeability and dyke direction: The treatment of anisotropic permeability requires further clarification and revision. In the rebuttal, the authors explain that anisotropy was introduced to limit lateral leakage, not to drive the flow. However, this important clarification is absent from the manuscript itself. Moreover, the influence of permeability anisotropy on dyke orientation is a nontrivial issue. Unless supported by a parametric study (e.g., varying anisotropy and boundary conditions), the manuscript should avoid making deterministic claims about dyke direction. The comment that anisotropic permeability in sedimentary rocks was not modeled also raises further questions: How many distinct permeabilities are present in the model? And how is anisotropy imposed? The sentence "Therefore, to limit leakage through the walls, we introduce an anisotropic permeability" requires clearer explanation and consistency with the modeling assumptions. References such as Rozhko et al. (2007), which demonstrate the role of external conditions in dyke orientation, should also be considered.

  **R:** We thank the reviewer for this helpful comment and agree that our rationale for using anisotropic permeability required clarification.

  The anisotropic permeability was introduced for a specific methodological purpose: to ensure the simulated dyke maintained a constant width, enabling a direct comparison with the one-dimensional poro-LEFM model. Its function was to prevent artificial widening, not to model the geological controls on dyke trajectory, a topic that falls outside the benchmarking scope of this paper.

  We have revised Section 2.2.2 to state this rationale explicitly, and we hope this clarifies our approach.

- Prescribed initial porosity and dyke emergence: The rebuttal asserts that the dyke is not prescribed but emerges from the PDEs given a specific initial porosity distribution. However, because this distribution is carefully shaped (anisotropic Gaussian) and placed at the base, it still strongly preconditions dyke formation. This explanation must be included in the manuscript. Better yet, presenting a comparison with a case using isotropic or random porosity would provide more convincing evidence for the claim of emergent behavior.

  **R:** We have revised the manuscript to clarify the role of the initial condition. The new text in Section 2.2.4 now explains that while the initial porosity field is intentionally designed to precondition the dyke's location and orientation such that it resembles the poro-LEFM model in Fig 1, the dyke itself — as a result of plastic failure — is an emergent feature that results dynamically from the governing equations.

- Mode-I vs. mode-II fracture and failure criterion (Line 296–299): The authors acknowledge that isolating mode-I fracture is not possible with their yield criterion, which incorporates both shear and compaction stresses. This important caveat is only included in the rebuttal and should be explained in the manuscript. This needs to be explained in the manuscript, not only in the response to reviewers.

  **R:** We thank the reviewer for this important point. As suggested, we have added a sentence to the manuscript (Section 3.1) to clarify that our yield criterion incorporates both shear and tensile failure and therefore does not isolate for pure Mode-I fracture.

- Terminology – Overpressure and dilatancy pressure (Line 526): The use of terms like "overpressure" and "dilatancy pressure" is potentially confusing. "Overpressure" typically refers to fluid pressure exceeding hydrostatic pressure, not to a generic pressure difference. If the authors choose to use this term, they must define it explicitly upon first use. A clearer solution would be to consistently refer to $\Delta P$ as "pressure difference" throughout the manuscript. Likewise, the term "dilatancy pressure" should be introduced more carefully to distinguish it from similar terms in existing literature.

  **R:** We thank the reviewer for highlighting this potential for confusion. As suggested, we have revised the manuscript to explicitly define "overpressure" upon its first use and have been more careful in our introduction of the term "dilatancy pressure" to clarify its specific role within our formulation.

- Constitutive modeling – Equations (D7) and (D8): The derivation of Equations (D7) and (D8) from the general viscoplastic framework (Equation C7) is still unclear. While the rebuttal provides some helpful interpretation, these explanations must be moved into the manuscript for the benefit of all readers. Clarifying these equations would significantly enhance the rigor of Appendix D.

  **R:** We thank the reviewer for highlighting this lack of clarity. We have revised Appendix D2 to better explain the decomposition of the viscoplastic work rate. The updated text now clarifies two key aspects: firstly, it explicitly states that the purpose of this subsection is to justify our choice of a small $\eta^{\mathrm{K}}$. Secondly, it clarifies the origin of Equation (D7), explaining that it is a definitional decomposition and not derived from Equation (C7). We hope this revised explanation makes our justification clearer.

---

## Author Response (AR3)

Dear Editors,

Thank you for your positive feedback of our revised manuscript, "Models of buoyancy-driven dykes using continuum plasticity and fracture mechanics: a comparison".

As requested, we have now incorporated references to the supplementary material within the manuscript. Specifically, we have made two additions:

- An in-text citation has been added to Section 3.1 (page 13), which now reads: "Figure 3(a) shows a snapshot of the porosity field from a representative numerical solution (see the video in the Supplementary Material)."

- A new "Video supplement" section has been included on page 21 to introduce the animation.

Thank you again for your advice.

Sincerely,

Yuan Li & Co-authors